# *ELF5* is a potential respiratory epithelial cell-specific risk gene for severe COVID-19

Maik Pietzner [1,2,20] ✉, Robert Lorenz Chua[3,20], Eleanor Wheeler [2], Katharina Jechow[3], Julian D. S. Willett[4,5], Helena Radbruch [6], Saskia Trump[7], Bettina Heidecker[8], Hugo Zeberg [9,10], Frank L. Heppner [6,11,12], Roland Eils[3,13,14], Marcus A. Mall [14,15,16], J. Brent Richards [4,5,17,18], Leif-Erik Sander [19], Irina Lehmann[7,14], Sören Lukassen [3], Nicholas J. Wareham [2], Christian Conrad [3,21] ✉ & Claudia Langenberg [1,2,21] ✉

Despite two years of intense global research activity, host genetic factors that predispose to a poorer prognosis of COVID-19 infection remain poorly understood. Here, we prioritise eight robust (e.g., ELF5) or suggestive but unreported (e.g., RAB2A) candidate protein mediators of COVID-19 outcomes by integrating results from the COVID-19 Host Genetics Initiative with population-based plasma proteomics using statistical colocalisation. The transcription factor ELF5 (*ELF5*) shows robust and directionally consistent associations across different outcome definitions, including a >4-fold higher risk (odds ratio: 4.88; 95%-CI: 2.47–9.63; p-value < 5.0 × 10⁻⁶) for severe COVID-19 per 1 s.d. higher genetically predicted plasma ELF5. We show that *ELF5* is specifically expressed in epithelial cells of the respiratory system, such as secretory and alveolar type 2 cells, using single-cell RNA sequencing and immunohistochemistry. These cells are also likely targets of SARS-CoV-2 by colocalisation with key host factors, including *ACE2* and *TMPRSS2*. In summary, large-scale human genetic studies together with gene expression at single-cell resolution highlight *ELF5* as a risk gene for severe COVID-19, supporting a role of epithelial cells of the respiratory system in the adverse host response to SARS-CoV-2.

[1]Computational Medicine, Berlin Institute of Health (BIH) at Charité – Universitätsmedizin Berlin, Berlin, Germany. [2]MRC Epidemiology Unit, University of Cambridge, Cambridge, UK. [3]Center for Digital Health, Berlin Institute of Health (BIH) at Charité – Universitätsmedizin Berlin, Berlin, Germany. [4]McGill Genome Centre, McGill University, Montréal, QC, Canada. [5]Lady Davis Institute, Jewish General Hospital, Montréal, QC, Canada. [6]Department of Neuropathology, Charité – Universitätsmedizin Berlin, corporate member of Freie Universität Berlin und Humboldt-Universität zu Berlin, Berlin, Germany. [7]Molecular Epidemiology Unit, Center for Digital Health, Berlin Institute of Health (BIH) at Charité – Universitätsmedizin Berlin, Berlin, Germany. [8]Department of Cardiology, Charité – Universitätsmedizin Berlin, corporate member of Freie Universität Berlin und Humboldt-Universität zu Berlin, Berlin, Germany. [9]Max Planck Institute for Evolutionary Anthropology, Leipzig, Germany. [10]Department of Neuroscience, Karolinska Institutet, Stockholm, Sweden. [11]Cluster of Excellence, NeuroCure, Berlin, Germany. [12]German Center for Neurodegenerative Diseases (DZNE) Berlin, Berlin, Germany. [13]Health Data Science Unit, Heidelberg University Hospital and BioQuant, Heidelberg, Germany. [14]German Center for Lung Research (DZL), associated partner site, Augustenburger Platz 1, 13353 Berlin, Germany. [15]Department of Pediatric Respiratory Medicine, Immunology and Critical Care Medicine, Charité-Universitätsmedizin Berlin, corporate member of Freie Universität Berlin and Humboldt-Universität zu Berlin, Berlin, Germany. [16]Berlin Institute of Health at Charité – Universitätsmedizin Berlin, Berlin, Germany. [17]Departments of Medicine, Human Genetics, Epidemiology, Biostatistics and Occupational Health, McGill University, Montréal, QC, Canada. [18]Department of Twin Research, King's College London, London, United Kingdom. [19]Department of Infectious Diseases and Respiratory Medicine, Charité - Universitätsmedizin Berlin, corporate member of Freie Universität Berlin, Humboldt-Universität zu Berlin, and Berlin Institute of Health (BIH), Berlin, Germany. [20]These authors contributed equally: Maik Pietzner, Robert Lorenz Chua. [21]These authors Jointly Supervised this work: Christian Conrad, Claudia Langenberg. ✉ e-mail: maik.pietzner@bih-charite.de; christian.conrad@bih-charite.de; claudia.langenberg@bih-charite.de

The COVID-19 pandemic, caused by the coronavirus SARS-CoV-2, has overwhelmed health care systems all over the world and caused >6.1 million deaths. The unprecedented pace of vaccine development, approval, and administration[1], has strongly reduced hospitalisations and prevented hundreds of thousands of deaths[2], and only achieving population-wide immunity will end the pandemic. However, it is unclear how long immunisation from vaccines or natural infection will last[3] and hospitalisation and death tolls remain high due to various factors, including the evolution of novel SARS-CoV-2 variants[4–6] and the missing availability of vaccines in low- and middle-income countries[7], requiring persistent efforts to identify host factors that predispose to poor outcomes.

So far, older age, male sex, smoking, obesity, social deprivation, ethnicity, and a high burden of pre-existing conditions have been consistently identified as risk factors for a poor prognosis among COVID-19 patients[8–11]. However, a severe disease course, including hospitalisations and fatal outcomes also occurs in otherwise low-risk patients. Further, the biology underlying disease progression and fatal outcomes remain largely unknown, with observational studies unable to dissect cause from consequence. Common variation in the human genome has now been robustly linked to a higher susceptibility to severe COVID-19 outcomes[12–15], offering novel and orthogonal insights to deep molecular profiling studies in patients employing single-cell sequencing or immunoprofiling[16–21]. For example, common variants at 3p21.31 (possibly mapping to *LZTFL1*[22] or *SLC6A20*[23]) or 12q24.13 (likely *OAS1*[24,25]) confer a 30–110% higher risk for severe outcomes of COVID-19[12–15], with suggested roles in alveolar type 2 (AT2) cells[17,22] or modulation of the host immune response[24,26]. However, a major obstacle to the clinical translation of these findings is the identification of the causal genes through which risk loci mediate their effect. Further, incorporating gene or protein expression quantitative trait loci (QTLs) via statistical colocalisation or Mendelian randomisation can highlight additional candidate genes[24,27–30]. Both techniques make use of a causal chain of events. Firstly, alleles are allocated at random at conception providing the opportunity to use them as instruments for causal inference. Secondly, genetic variants near protein-encoding loci (cis-pQTLs) that associate with protein levels in healthy individuals before viral exposure can serve as instruments for lifelong exposure to higher or lower protein levels. Therefore, establishing that the same genetic signal associates with protein levels and a poor prognosis of COVID-19 provides strong evidence for a causal role of the protein in the aetiology of the disease. This

is particularly relevant for COVID-19, which is characterised by a hyperimmune response and a profound impact on the plasma proteome[31] limiting insights from ad hoc cross-sectional proteomic studies. Such genetically informed strategies have already identified potential druggable targets, including *ACE2*[27], or modulators of the immune response such as OAS1[24].

Here, we exploit a proteome-wide colocalisation screen, incorporating cis-pQTLs for >1100 protein targets across two different platforms[30,32], to identify proteomic modulators of SASRS-CoV-2 infection and COVID-19 prognosis using genome-wide summary statistics for four different outcome definitions (ranging from susceptibility to severity) as released by the COVID-19 Host Genetics Initiative (HGI)[15] (https://www.covid19hg.org/, release 6) (Fig. 1). We demonstrate the ability of genetically informed plasma proteomics to identify causal genes and proteins for severe COVID-19 and pinpoint the responsible cell types through the integration of orthogonal data.

## Results

### Putative protein mediators of disease risk

We identified a total of 8 putative candidate causal proteins distributed across three tiers of confidence by systematically testing for a shared genetic signal at protein-coding loci (±500 kb) across 1121 protein targets (see Methods) and COVID-19 outcomes (Supplementary Table 1) using statistical colocalisation (Supplementary Data 1–5, Methods). We replicated with high confidence proteins encoded at *ABO* and *OAS1*[24,27,28], refined with robust confidence proteins encoded at loci near *ELF5* and *SFTPD*[15], and further provide suggestive evidence that proteins encoded at *CSF3, HSP40, RAB2A, and NUDT5* might modulate SARS-CoV-2 susceptibility or the course of COVID-19 (Table 1). All findings were virtually unaffected when systematically varying prior settings in colocalisation analysis (Supplementary Data 6). Further, all genetic variants possibly linking proteins to COVID-19 had the highest effect estimates for severe COVID-19 (Supplementary Data 5), although a more sophisticated Bayesian approach as reported by the COVID-19 HGI classified variants at *ABO* to be most likely related to infectious susceptibility rather than severe COVID-19[15]. This implies that the encoded protein, Histo-blood group ABO system transferase (BGAT), may somehow contribute to a higher susceptibility of infection but not a poorer prognosis.

Of the 6 refined or suggestive candidates, ELF5 showed significant and directionally consistent associations across all four COVID-19

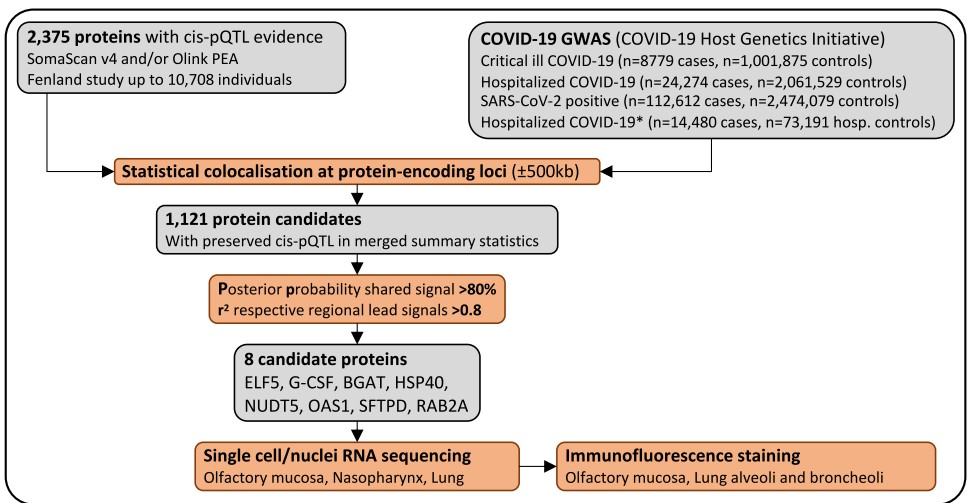

**Fig. 1 | Flowchart of the study design.** We tested whether the same genetic signal that was associated with higher/lower protein abundances was also associated with higher/lower risk for COVID-19 based on cis protein quantitative trait loci (cis-pQTLs). We further investigated the expression of targeted proteins/genes using single-cell and single nuclei RNA sequencing in samples of the respiratory system and confirmed the expression of the most robust candidate ELF5 using immuno-fluorescence staining.

## Table 1 | Summary of candidate protein prioritisation across four different COVID-19 outcomes

| Protein name | Plt | Gene | outcome | LD* | colocalisation PP | | rsid | EA | NEA | EAF | Protein | | | COVID-19 | | |
|---|---|---|---|---|---|---|---|---|---|---|---|---|---|---|---|---|
| | | | | | H3 | H4 | | | | | Effect | SE | P value | Effect | SE | P value |
| **Tier 1 – genome-wide statistical significance (p < 5 × 10⁻⁸) in protein and COVID-19 summary statistics** | | | | | | | | | | | | | | | | |
| Histo-blood group ABO system transferase | S | ABO | B2 | 1.00 | 0.89% | 99.11% | rs505922 | t | c | 0.67 | -1.23 | 0.01 | <1E-300 | -0.100 | 0.01 | 2.09E-20 |
| 2-5'-oligoadenylate synthase 1 | S | OAS1 | B2 | 0.99 | 11.75% | 88.25% | rs6489868 | c | g | 0.64 | -0.08 | 0.01 | 5.94E-09 | 0.070 | 0.012 | 1.31E-09 |
| 2-5'-oligoadenylate synthase 1 | S | OAS1 | A2 | 0.92 | 12.55% | 87.44% | rs6489868 | c | g | 0.64 | -0.08 | 0.01 | 5.94E-09 | 0.115 | 0.019 | 2.06E-09 |
| **Tier 2 – genome-wide statistical significance (p < 5 × 10⁻⁸) in protein or COVID-19 summary statistics** | | | | | | | | | | | | | | | | |
| ETS-related transcription factor Elf5 | S | ELF5 | B2 | 1.00 | 0.47% | 99.07% | rs766826 | t | c | 0.36 | -0.07 | 0.01 | 5.38E-06 | -0.082 | 0.012 | 1.16E-11 |
| Pulmonary surfactant-associated protein D | O | SFTPD | B2 | 1.00 | 0.24% | 99.73% | rs721917 | g | a | 0.40 | -0.67 | 0.05 | 5.37E-31 | 0.055 | 0.01 | 2.08E-07 |
| 2-5'-oligoadenylate synthase 1 | S | OAS1 | C2 | 0.92 | 11.92% | 87.70% | rs6489868 | c | g | 0.64 | -0.08 | 0.01 | 5.94E-09 | 0.021 | 0.004 | 1.67E-06 |
| DnaJ homologue subfamily B member 1 | S | DNAJB1 | A2 | 0.98 | 1.68% | 93.62% | rs35258260 | a | g | 0.85 | -0.11 | 0.02 | 4.61E-10 | -0.105 | 0.027 | 1.29E-04 |
| ADP-sugar pyrophosphatase | S | NUDT5 | B2 | 1.00 | 1.43% | 86.27% | rs7895525 | t | c | 0.76 | -0.11 | 0.02 | 3.29E-12 | 0.047 | 0.012 | 1.34E-04 |
| Ras-related protein Rab2A | S | RAB2A | B2 | 0.98 | 11.98% | 84.01% | rs6995515 | a | c | 0.63 | -0.07 | 0.01 | 2.57E-08 | -0.041 | 0.01 | 1.47E-04 |
| **Tier 3 – suggestive statistical significance (5 × 10⁻⁸ < p < 1 × 10⁻⁵) in protein and COVID-19 summary statistics** | | | | | | | | | | | | | | | | |
| ETS-related transcription factor Elf5 | S | ELF5 | B1 | 1.00 | 0.67% | 96.68% | rs766826 | t | c | 0.36 | -0.07 | 0.01 | 5.38E-06 | -0.078 | 0.017 | 4.01E-06 |
| ETS-related transcription factor Elf5 | S | ELF5 | A2 | 1.00 | 0.75% | 97.93% | rs766826 | t | c | 0.36 | -0.07 | 0.01 | 5.38E-06 | -0.105 | 0.023 | 5.05E-06 |
| Granulocyte colony-stimulating factor | O | CSF3 | C2 | 0.88 | 6.04% | 88.20% | rs3826331 | c | t | 0.62 | -0.25 | 0.06 | 7.54E-05 | 0.019 | 0.004 | 5.47E-06 |

outcome = type of COVID-19 outcome; A2 = severe COVID-19 outcome; B1 = Hospitalised COVID-19 vs population; B2 = Hospitalised COVID-19 vs non-hospitalised COVID-19; C2 = SARS-CoV-2 positive vs population; Plt = Platform: S = SomaScan v4 or O – Olink PEA; PP = posterior probability; H3 = distinct genetic signals; H4 = shared genetic signal; LD* = linkage disequilibrium between respective regional lead genetic variants for protein levels and COVID-19 outcomes. Effect estimates, standard errors (SE), and p values from linear regression models are shown for candidate genetic variants and were obtained from Pietzner et al.[30,32] for protein levels and the COVID-19 HGI for COVID-19 outcomes[15].

outcomes included as part of the COVID-19 HGI[15], with the strongest effects for more severe outcomes (Fig. 2 and Supplementary Data 5) and strong evidence for a shared genetic signal for outcomes indicating a poorer prognosis (hospitalisation and severe COVID-19; Posterior probability (PP) > 80%). A definition in line with the classification by the COVID-19 HGI[15]. For example, a 1 s.d. increase in genetically predicted ELF5 plasma abundances was associated with an almost fivefold higher risk for severe COVID-19 (odds ratio: 4.88; 95% CI: 2.47–9.63; $p$ value < $5.0 \times 10^{-6}$) in single-instrument Mendelian randomisation (MR) analysis using the lead cis protein quantitative trait locus (cis-pQTL) as the genetic instrument (Fig. 2, see Methods).

The remaining candidate proteins showed at least suggestive evidence for selected COVID-19 outcomes in our initial analysis (Fig. 2), with G-CSF showing medium support (regional PP = 64%) for a shared genetic signal with greater susceptibility to infection and severe COVID-19, using multi-trait colocalisation[33] (Supplementary Fig. 1).

### ELF5 is the candidate causal gene at 11p13 for severe COVID-19

The lead cis-pQTL for ELF5, rs766826 (MAF = 35.9%), has been reported by the COVID-19 HGI[15] and is further in strong linkage disequilibrium (LD; $r^2 = 0.81$) with a recently identified variant rs61882275 associated with severe COVID-19 in an independent study using whole genome sequencing[34]. The causal gene, however, remained ambiguous, since *ELF5* and *CAT* were reported as putative causal genes at this locus based on evidence from gene expression studies for rs766826[35]. Both encoded protein products are captured by our proteomic data allowing us to test for the causal gene across multiple molecular layers. We prioritised *ELF5* as the causal gene at this locus through a cluster of colocalising phenotypes, including three different COVID-19 outcomes, *ELF5* expression in the lung, and ELF5 abundances in plasma (PP = 93% for a shared signal across all outcomes), with rs766826 being the most likely (PP = 99%) underlying causal variant (Fig. 3). We further tested for colocalisation of the cis-pQTL for ELF5 and gene expression across all GTEx tissues and identified colocalisation specific to expression in lung but not in other tissues, including tissues with high ELF5 expression such as breast, prostate, or salivary glands (Supplementary Fig. 2). This finding points towards a specific role of the major C-allele of rs766826 in increasing the expression of *ELF5* in the lung (beta = 0.24, $p$ value = $5.3 \times 10^{-15}$) with subsequent higher risk for severe COVID-19 (odds ratio = 1.11, 95% CI:1.06–1.16; $p$ value < $5.0 \times 10^{-6}$) and higher abundances of ELF5 in blood (beta = 0.07, $p$ value < $5.4 \times 10^{-6}$), the latter likely via cell turnover or injury of lung tissue since ELF5 is not predicted to be actively secreted into blood[36].

While we identified strong evidence for a shared genetic signal (PP > 80%) between the lead cis-pQTL for the other candidate causal gene at this locus (*CAT*) and the corresponding cis-eQTL in 29 out of 49 tissues in GTEx v8, indicating convergence of gene and protein expression, plasma levels of catalase (encoded by *CAT*) did not colocalise with COVID-19 phenotypes. We further observed that *CAT* expression in lung and whole blood formed a separate cluster (PP = 56.6%) at the same genetic locus possibly explained by rs35725681 (PP = 36.8%, Supplementary Fig. 3).

### rs766826 resides in an open chromatin region in the lung and is associated with lung function

While *ELF5* is known to be highly expressed in multiple tissues, we demonstrated that the genetic signal shared between ELF5, severe COVID-19, and gene expression was specific to the lung. We observed that rs766826 mapped to an open chromatin region in AT2 cells from the lung but not in other tissues with high expression of *ELF5*[37], such as the mammary gland, prostate, or kidney (Supplementary Fig. 4), providing a potential explanation for the tissue-specific effect. We further observed strong evidence that rs766826 is the most likely candidate causal variant across ancestries based on high posterior inclusion

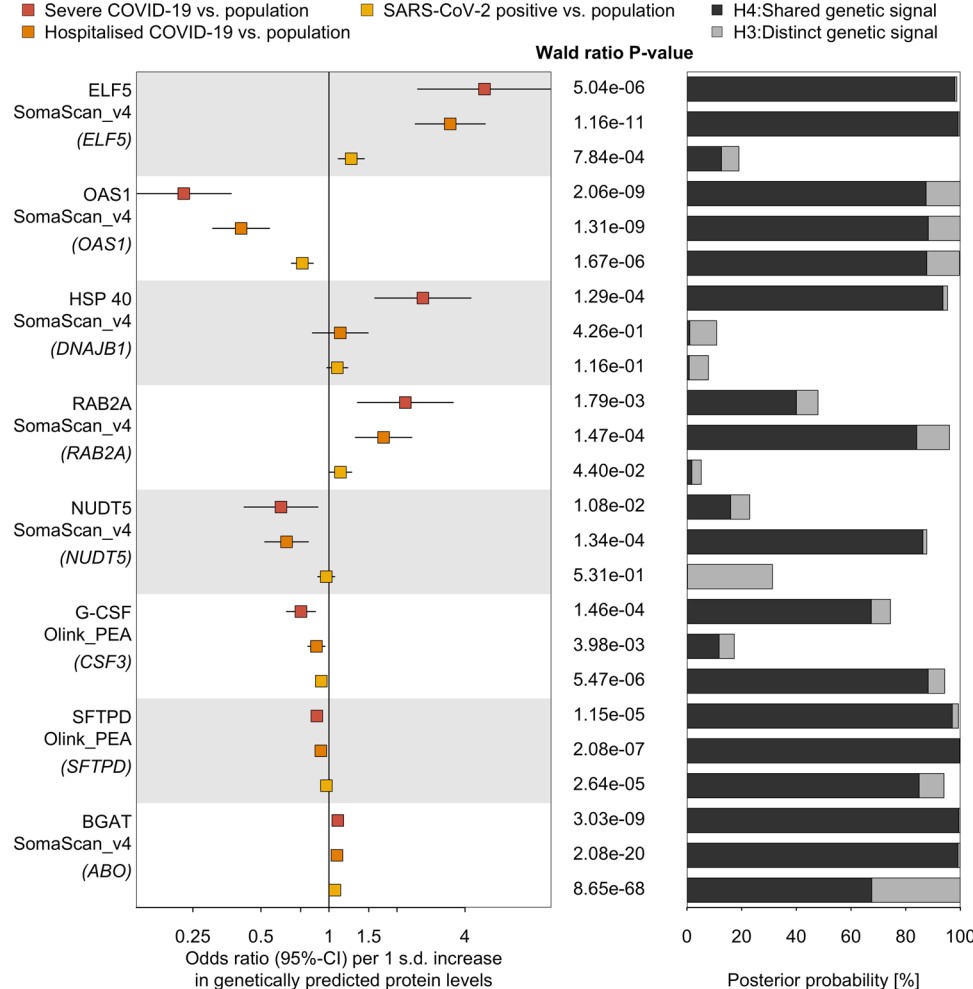

**Fig. 2 | Proteins genetically linked to various COVID-19 outcomes.** Odds ratios (rectangles) and 95% CIs (lines) for the genetically predicted effect of protein levels on three different outcome definitions and control populations for COVID-19 (left), including protein targets with strong evidence for statistical colocalisation for at least one definition (right). Proteins are ordered by absolute effect size for severe COVID-19. The column in the middle reports the *p* values for Wald ratio estimates from a single variant Mendelian randomisation analysis. Stacked bar charts represent the posterior probability that protein levels and COVID-19 outcomes share a genetic signal in a 500 kb flanking region of the protein-encoding gene (black) or rather represent two distinct signals (grey). For each protein, the proteomics platform and the protein-encoding gene (in brackets) are listed. ELF5 ETS-related transcription factor Elf5, OAS1 2′–5′-oligoadenylate synthase 1, HSP40 DnaJ homologue subfamily B member 1, RAB2A Ras-related protein Rab2A, NUDT5 ADP-sugar pyrophosphatase, SFTPD Pulmonary surfactant-associated protein D, G-CSF Granulocyte colony-stimulating factor, BGAT Histo-blood group ABO system transferase. The data underlying this figure is given in Supplementary Data 5. Sample sizes for COVID-19 outcomes can be found in Supplemental Table 1; 10,708 and 485 participants were included for SomaScan and Olink analysis, respectively.

probabilities (PIP) in European (PIP = 66%; 17,992 cases and 1,810,493 controls) and African individuals (PIP = 83%; 2113 cases and 121925 controls) when performing stepwise fine-mapping using European PIPs as priors for the smaller African data set (Supplementary Fig. 5, see Methods).

To systematically test phenotypic consequences of rs766826 or proxies in strong LD ($r^2 > 0.8$), we queried the OpenGWAS database[38] and performed colocalisation for all suggestive associations observed at $p < 10^{-4}$. The only association with evidence for a shared genetic signal with ELF5 expression and COVID-19 outcomes was seen for lung function (PP = 99%) (Supplementary Fig. 6). However, effect directions were somewhat counterintuitive, since the ELF5-increasing and COVID-19 risk C-allele was associated with better lung function (beta = 0.01, *p* value < $1.6 \times 10^{-5}$) based on the quotient between forced expired volume in 1 second (FEV$_1$) and forced vital capacity (FVC) from spirometry in population-based, that is, COVID-19 free studies[39]. A similar phenomenon has been described for rs35705950 mapped to *MUCB5* and risk for idiopathic pulmonary fibrosis, a condition that shares commonalities with severe COVID-19[40].

**ELF5 is expressed in epithelial cells of the respiratory system**
Motivated by the supposedly lung-specific genetic signal for ELF5 and the general relevance of the entire respiratory system for severe COVID-19, we reanalysed single-cell and single nucleus RNA sequencing (scnRNAseq) data sets generated by us[41–43] and others[44] across different sites from healthy donors (Fig. 4, see Methods). We observed that *ELF5* was almost exclusively expressed by different epithelial cells of the respiratory system (Fig. 4). Its expression pattern was shared with the viral entry receptor *ACE2* and associated proteases, such as *TMPRSS2*. Specifically, sustentacular and Bowman gland cells from the olfactory mucosa and secretory epithelial cells from the pseudostratified epithelium of the nasophar-ynx and distal bronchioles of the lung showed high expression levels of *ELF5* (Fig. 4C). They similarly expressed host proteins utilised by SARS-CoV-2, including *ACE2* or *TMPRSS2*, suggesting that putative target cells of SARS-CoV-2 express high quantities of *ELF5*.

Lungs from deceased COVID-19 patients consistently show signs of massive alveolar damage[17,45]. While *ELF5* expression was

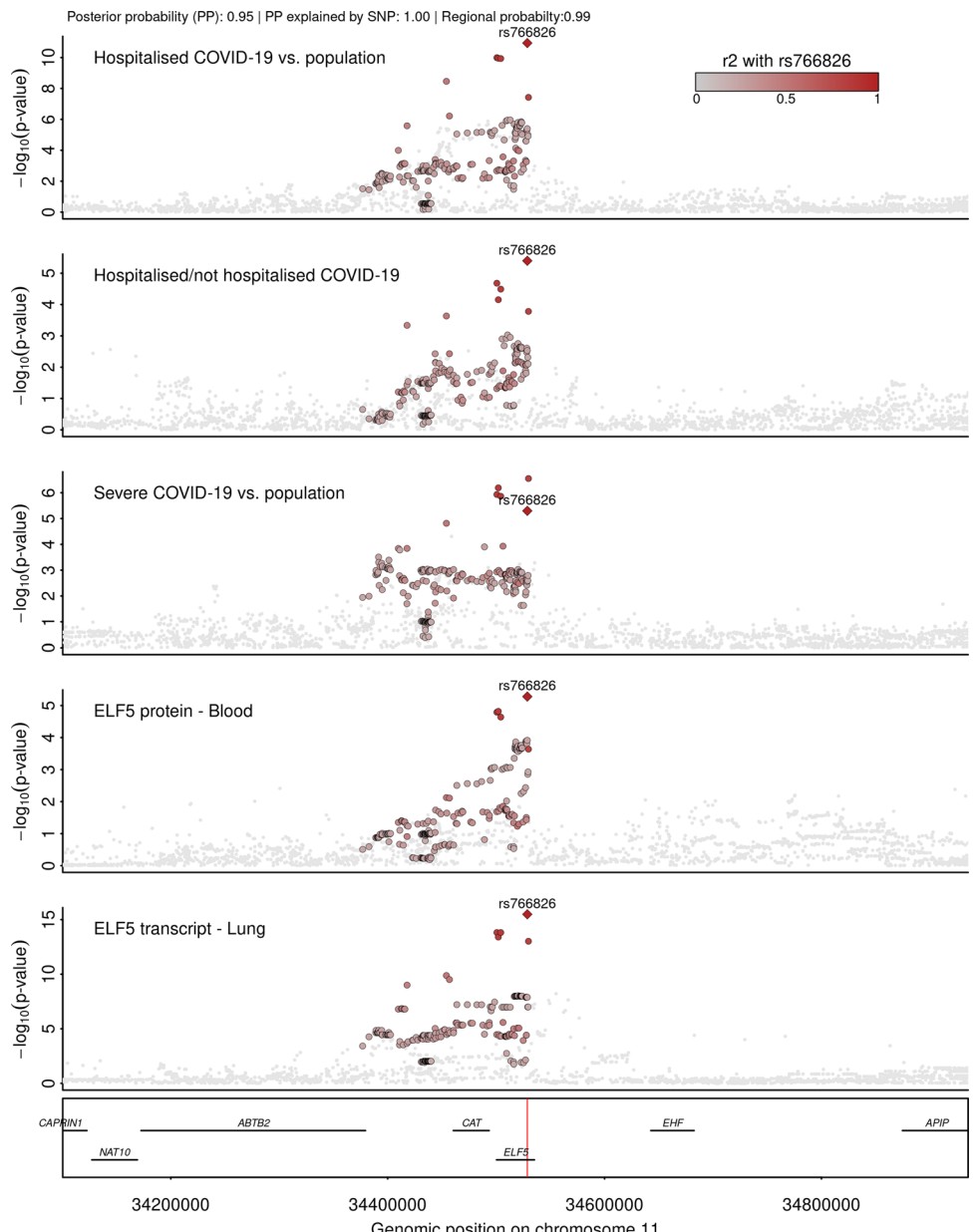

**Fig. 3 | Stacked regional association plots at *ELF5*.** Each panel contains regional association statistics from linear regression analysis (*p* values) for the trait listed in the upper left corner along genomic coordinates. Each dot represents a single nucleotide polymorphism and colours indicate linkage disequilibrium (LD; *r²*) with the most likely causal variant (rs766826) at this locus (darker colours stronger LD). The position of rs766826 in the genome is highlighted by a red line on the lowest panel. Summary statistics for COVID-19-related outcomes were obtained from the COVID-19 HGI[15], for protein abundances from Pietzner et al.[30], and gene expression in lungs from GTEx version 8[35]. LD was calculated based on 8350 unrelated white-British participants of the Fenland cohort.

highest in secretory cells, AT2 but not AT1 cells showed consistent expression of *ELF5* in our lung data set of COVID-19-free donors[42] (Fig. 4). This finding coincided with classical lineage markers for AT2 cells such as *SFTPD*, which was also highly expressed in mitotic cells (Fig. 4C). We note that our proteogenomic screen prioritised *SFTPD* as a candidate gene for severe COVID-19 in line with other studies[15,34], leaving the possibility of potential interactions between candidate mediators. AT2 cells not only co-express SARS-CoV-2 host factors, including *ACE2*[42,46,47], but they also fulfil key roles in maintaining normal lung function by producing surfactants that reduce surface tension and prevent alveolar collapse[48], and further serve as resident stem cells essential for maintenance and repair of the alveolar epithelium[49].

**Immunofluorescence staining validates scnRNAseq results and shows high ELF5 expression in AT2 cells of postmortem COVID-19 samples**

We validated the expression of ELF5 at the protein level in the different epithelial cells of the olfactory mucosa and lungs using immuno-fluorescence staining in non-COVID-19 samples (Fig. 5 and Supplementary Figs. 7, 8). Within the olfactory mucosa, sustentacular cells (KRT18⁺) and horizontal basal cells (KRT18⁻) of the olfactory epithelium (above the dashed line, Fig. 5A) and the bowman gland cells (KRT18⁺) within the lamina propria (below the dashed line, Fig. 5A) were positively stained for ELF5 (Fig. 5A and Supplementary Fig. 7A). We further validated protein expression of ELF5 in AT2 (SFTPC⁺) and epithelial cells (EPCAM⁺) of the airways in lung consistent with the

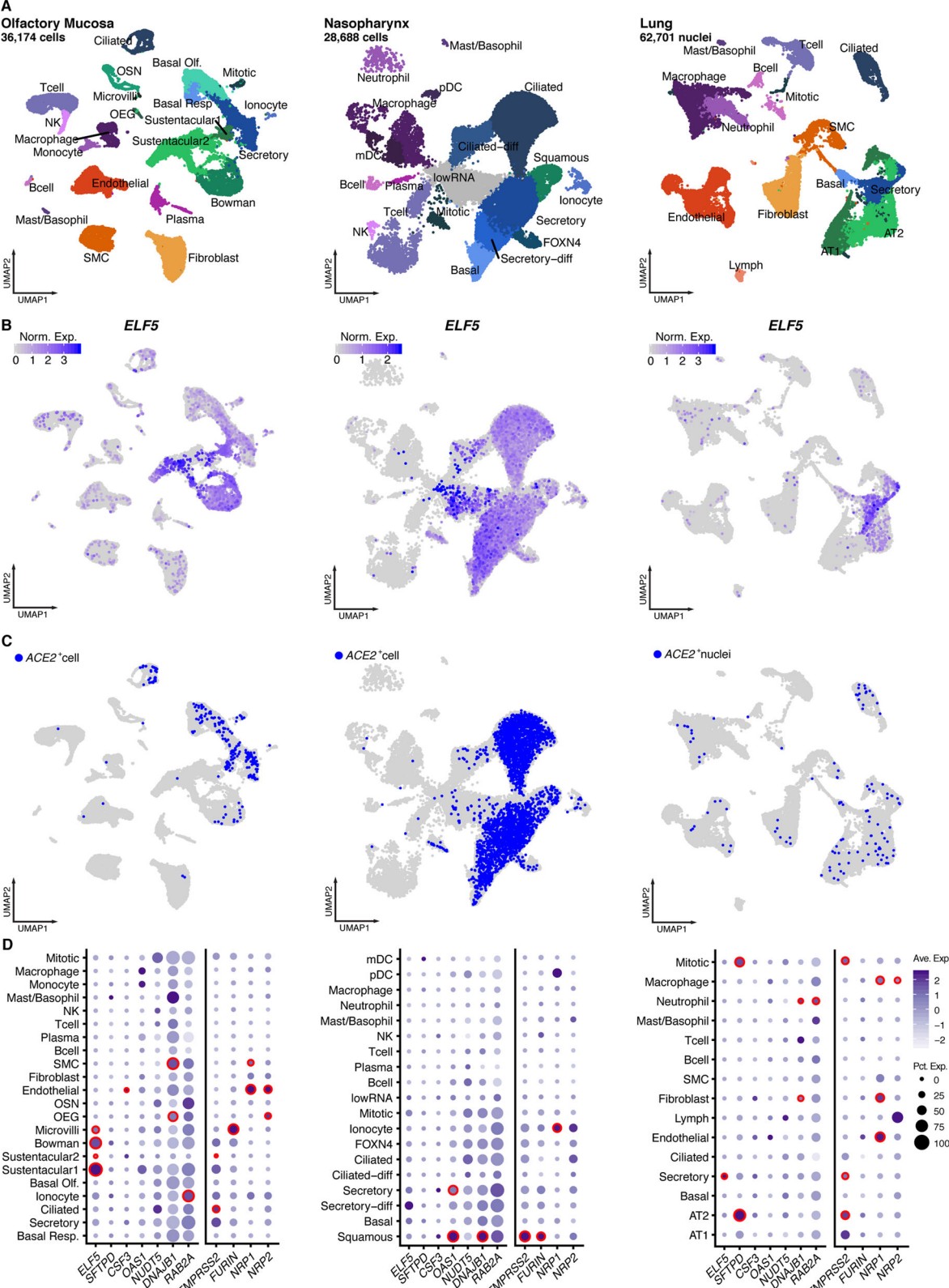

**Fig. 4 | Expression of candidate genes in different single-cell data sets covering the respiratory system. A** UMAP representation of single-cell/nuclei RNA sequencing data from three data sets (olfactory mucosa, nasopharynx, and lung) with annotations of cell types. **B** Expression levels of *ELF5* across all cells identified in all three data sets. **C** *ACE2*⁺ cells in all three data sets. **D** Dot plots showing the number of cells positive for candidate genes (size). The colour gradient indicates scaled average expression levels and red frames indicate significantly higher expression (MAST-based test; false discovery rate adjusted *p* value < 0.05) of the target gene in one cell type compared to all others. Expression patterns for suggested host factors required for viral entry, such as *TMPRSS2, FURIN, NRP1*, as well as *NRP2*, have been added as a comparator. Data have been obtained from Lukassen et al.[42], Loske et al.[43], Gassen et al.[41], and Durante et al.[44].

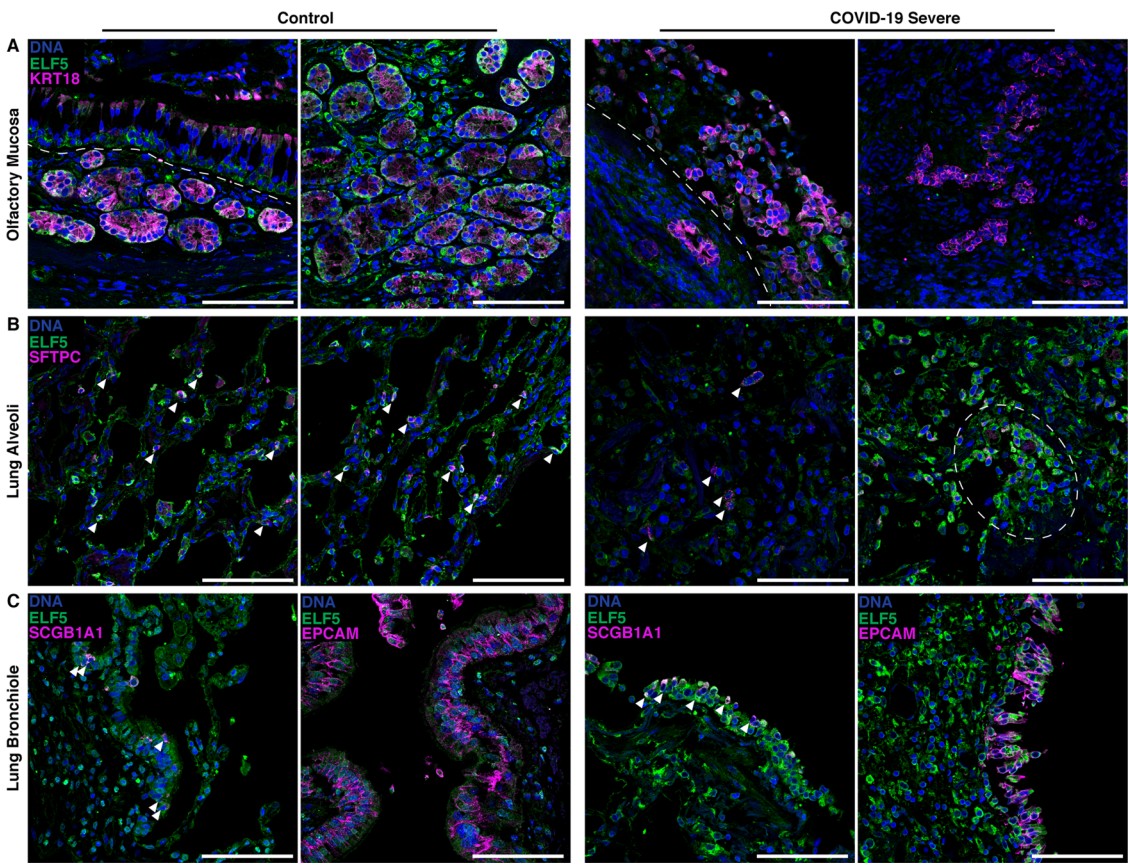

**Fig. 5 | ELF5 expression by epithelial cells of the olfactory mucosa and lung.**
Immunofluorescent staining of ELF5 in control and COVID-19 patients in the
**A** olfactory mucosa, **B** lung alveoli, and **C** lung bronchiole. **A** Dashed lines separate
the olfactory epithelium and the lamina propria. **B** Arrowheads highlight AT2 cells
expressing ELF5; dashed outline highlights clusters of AT cells expressing ELF5.

**C** left: epithelial cells expressing ELF5; right: arrowheads highlight airway epithelial
cells expressing ELF5. Marker genes for sustentacular and Bowman gland cells
(**A** KRT18), alveoli type II cells (**B** SFTPC), pan-epithelial cells (**C** EPCAM), and
secretory cells (**C** SCGB1A1) are shown in purple. Validation staining for each tissue:
control ($n = 2$); COVID-19 ($n = 2$). Scale bar = 100 μm.

scnRNAseq experiments (Fig. 5B, C and Supplementary Fig. 7B–D). We
observed similar validation of scnRNASeq experiments by immuno-
fluorescence staining for ACE2 and TMPRSS2 (Supplementary Fig. 9).

We next investigated ELF5 expression within the same tissues in
samples from two patients who rapidly died from severe COVID-19
within ≤14 days (Fig. 5A–C). We observed an injured olfactory epi-
thelium, including highly disrupted Bowman glands and very few
cells showing ELF5 expression (Fig. 5A, left; Supplementary Fig. 7A),
and further loss of structural integrity of the alveolar region of the
lung with only very few AT2 cells within the damaged region.
However, AT2 cells characterised by high ELF5 expression also
formed clusters, possibly reflective of their activated state to
regenerate the epithelia[49,50] (Fig. 5B and Supplementary Fig. 7B).
Similarly, secretory cells (SCGB1A1+) along with other epithelial cells
of the airway mucosa (e.g., ciliated and basal cells) expressed ELF5
and were also injured over the course of SARS-CoV-2 infection
(Fig. 5C, Supplementary Fig. 7C, D). These observations suggest that
*ELF5* expression might play a dynamic role during SARS-CoV-2
infection and COVID-19.

We finally observed potential signs of remodelling and high ELF5
expression in postmortem respiratory tissue samples of two COVID-19
patients with a fatal but not rapid disease course due to intensive
treatment, including extracorporeal membrane oxygenation
(≥14 days, termed as 'later death'; Supplementary Fig. 10A). Mucosal
structures were similar to controls, although structural integrity was
not fully restored. We further observed potentially regenerative
structures with either AT2 cells or airway epithelial cells highly
expressing ELF5 that may indicate an active wound healing

response[50–52] (Supplementary Fig. 10B, C). Delorey et al. recently
showed the induction of a regenerative programme in cells of the
airway and alveolar epithelium after SARS-CoV-2 infection[17]. However,
AT2 cell renewal and AT1 cell differentiation were inhibited in COVID-
19, leading to an accumulation of cells in this regenerative transitional
cell state and potentially lung failure[17].

### *ELF5* and *TMPRSS2* are co-expressed

To derive a possible hypothesis of how *ELF5* expression might be
linked to severe COVID-19, we collated a list of candidate genes, that
were either regulated by or co-expressed with *ELF5* (see Methods and
Supplementary Data 7). Among the set of candidate genes were mul-
tiple members of the transmembrane serine protease family, including
*TMPRSS2* and *TMPRSS4*, which have been shown to be essential for viral
entry by priming of the spike protein[53,54]. We observed a positive cor-
relation between *ELF5* and *TMPRSS2* expression in sustentacular cells
($r = 0.15$, $p < 4.5 \times 10^{-6}$, Supplementary Fig. 11), the cell type with the
highest *ELF5* expression in scnRNAseq data sets, but no correlation
with *TMPRSS4* expression. The correlation with *TMPRSS2* expression
was also above the 95th percentile of correlation coefficients across all
genes. Further, genes highly correlated with *ELF5* expression were also
significantly (enrichment score 0.31; $p = 0.013$) enriched among col-
lated target genes, minimising the possibility of a measurement arte-
fact. While such correlation analysis using single-cell data must be
treated with caution, overexpression of *Efl5* in a mouse model showed
a three-fold increase in *Ace2* and a two-fold higher expression of
*Tmprss4* in AT2 cells[55], providing additional evidence that *ELF5* might
be involved in the regulation of key host factors for SARS-CoV-2.

To formally test for pathways associated with *ELF5* expression, we performed cell-type specific pathway enrichment analysis using the collated set of putative ELF5 targets or co-expressed genes and observed a consistent enrichment of biosynthetic pathways like mRNA and peptide processing, and possibly among genes involved in epithelial barrier formation (Supplementary Fig. 12). The latter aligns with a substantial impact of *Elf5* overexpression on the differentiation of the lung epithelium in mouse models of lung development, leading to dilation of the airways[55].

### Drug target identification

We queried all identified candidate proteins in the Open Targets database[56] to identify repurposing opportunities for COVID-19. While none of the proteins had already been approved drugs or drugs in clinical trials, recombinant human G-CSF (rhG-CSF), such as filgrastim and lenograstim, is used to treat the neutropenia caused by chemotherapy to stimulate the production of granulocytes from the bone marrow[57]. In line with this, phenotypic associations identified in a phenome-wide colocalisation analysis at *CSF3* (±500 kb, encoding G-CSF) provided robust evidence for a shared genetic signal between plasma G-CSF and granulocyte and other white blood cell counts[58], with consistent positive associations across candidate genetic variants (Fig. 6 and Supplementary Data 8). This suggests the ability of the cis-pQTL to instrument the function of the protein and allowed testing of the potential effect of G-CSF supplementation for (severe) COVID-19 in silico. We observed a 25% (odds ratio: 0.75; 95% CI: 0.65–0.87; $p$ value $< 1.5 \times 10^{-4}$) reduction in the risk for severe COVID-19 per 1 s.d. higher genetically predicted G-CSF (Supplementary Data 5). A previous smaller, independent study that used a different proteomic technology observed directionally consistent results but did not reach statistical significance[59]. Together, these results provide in silico evidence that people with genetically higher plasma G-CSF abundances are less likely to develop (severe) COVID-19 and suggest that treatment with rhG-CSF might decrease the risk for symptomatic or even severe COVID-19. A recent randomised clinical trial[60] among 200 COVID-19 patients with pneumonia and severe lymphopenia observed a significantly lower number of patients developing critical illness when treated within the first three days of inclusion with 5 μg/kg rhG-CSF. However, leucocytosis was common in the treatment arm, including severe cases, which may limit the general application. For example, it might be conceivable that rhG-CSF treatment in COVID-19 patients with a strong immune response stimulates an adverse hyperinflammatory state and hence only a subgroup of COVID-19 patients might benefit.

## Discussion

Multiple host genetic variants have been identified[12–15] that predispose SARS-CoV-2 infected individuals to a severe course of COVID-19, including hospitalisation and risk of death, pointing to causal mechanisms. To translate these findings into clinical management or the identification of novel drug targets and repurposing opportunities, a deep understanding of the involved causal genes is needed. We identified six candidate causal genes and their proteins by refining known risk loci (*ELF5*, *SFTPD*) and by prioritising suggestive loci (*CSF3*, *RAB2A*, *HSP40*, *NUDT5*) through the integration of plasma proteomics. We demonstrate that the strongest and most robust candidate, *ELF5* (associated with a >4-fold higher risk to develop severe COVID-19), is specifically expressed in primary target cells of SARS-CoV-2 (for example, sustentacular[61], AT2[46], and secretory or ciliated epithelial cells[62]) with evidence of co-expression with genes encoding key host factors, such as *ACE2* and *TMPRSS2*, using scnRNAseq data across various sites of the respiratory system. We further find genetically anchored evidence that aligns with a recent clinical trial[60] suggesting human recombinant granulocyte colony-stimulating factor (G-CSF) as

a potential treatment option among patients with COVID-19 and severe lymphopenia to mitigate adverse outcomes.

*ELF5* is a member of the erythroblast transformation-specific (Ets) transcription factor family and is best known for its possible role in breast or prostate cancer, tissues with high fractions of epithelial cells[63,64], and less for its possible role in lung development[55,65] and possibly cystic fibrosis[66]. Experimental models to study the role of ELF5 are difficult since $Elf5^{-/-}$ mice are embryonic lethal[67]. However, the recent development of transgenic mouse models[68] and our scnRNAseq data provide strong evidence that *ELF5* is expressed in epithelial cells of the respiratory system of adult mice and humans. Early work in lung tissue cultures and mouse models described a dynamic expression pattern of *Elf5* during embryogenesis and lung branching, including almost complete downregulation in distal lung postnatally, while residual expression in proximal airways persisted[55,65]. Overexpression of *Elf5* during early but not late embryonal development (after E16.5) caused a severe cystic lung phenotype characterised by disrupted branching and a dilated airway epithelium[55], characteristics that are also seen in autopsies of COVID-19 patients[45]. While such a drastic intervention in mouse models is not comparable to the subtle effect of a common genetic variant, the observation that key host factors for SARS-CoV-2 (*Ace2* and *Tmprss4*) are upregulated in *Elf5*-overexpressing AT2 cells partly aligns with our observations using scRNAseq data.

The role of *ELF5* in secretory and AT2 cells of the airway and alveolar epithelium, respectively, may have potential implications to the wound healing response. As cells with stem-like capacity, they are involved in the maintenance and repair of their respective cellular niches[49,69]. Thus, any surviving secretory and AT2 cells that drive the repopulation of the epithelium could potentially have aberrant repair programmes mediated by *ELF5* and therefore possibly rs766826. An accumulation of AT2 cells in a regenerative transitional cell state has recently been suggested for COVID-19[17].

Up to 60% of COVID-19 patients report transient anosmia[70]. The underlying aetiology, however, remains largely elusive. Direct infection and hence damage of olfactory sensory neurons by SARS-CoV-2 could be one obvious explanation. Viral particles have been shown to be present in neuronal cells of the olfactory mucosa possibly presenting a route for CNS infection[71], however, the generally undetectable expression levels of *ACE2* in those cells make them an unlikely primary target compared to, for example, epithelial cells[61]. Previous studies suggested that the loss of essential supporting cells, sustentacular cells, in the olfactory mucosa causes anosmia[61,72]. Sustentacular cells have been suggested as primary targets of SARS-CoV-2 based on high *ACE2* expression[46,61,73], supported by in vivo models showing a high viral load and rapid desquamation of the olfactory epithelium following infection[74,75]. A finding in line with our observations from samples of COVID-19 patients. Our observation that sustentacular cells, as well as other secretory epithelial cells in the olfactory mucosa, express high levels of *ELF5* along with a possible link to *ACE2* expression, might indicate a possible modulating role of *ELF5* expression for this common symptom. However, a recent genome-wide association study (GWAS) for anosmia[76] among self-reported COVID-19 cases did not yet identify rs766826 and hence *ELF5* expression. Larger GWAS for anosmia and functional studies are needed to clarify a possible role of *ELF5* in the onset of anosmia during SARS-CoV-2 infection.

We provide genetically anchored evidence that people with higher plasma G-CSF abundances are less likely to develop severe COVID-19, suggesting a possible protective effect possibly via early recruitment of neutrophils to the entry sites of SARS-CoV-2[43]. Colony-stimulating factors, such as G-CSF, are haematopoietic growth factors and are actively investigated as treatment options for COVID-19[77]. A recent open-label, multicentre, randomised clinical trial[60] evaluated the efficacy of rhG-CSF to improve symptoms among 200 COVID-19 patients with lymphopenia (lymphocyte cell count <800 per μL) but

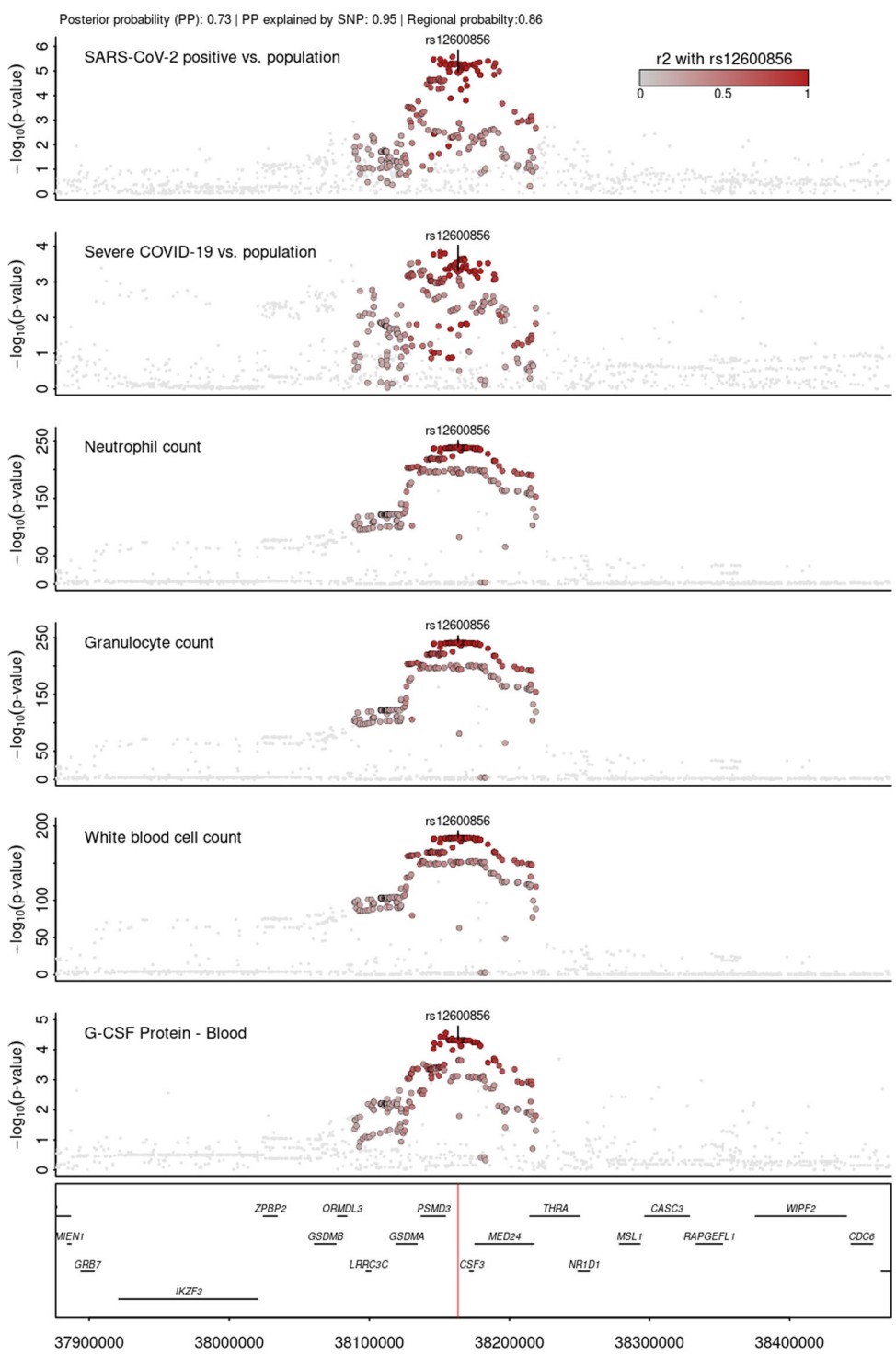

**Fig. 6 | Stacked regional association plots at *CSF3*.** Each panel contains regional association statistics (*p* values) from linear regression analysis for the trait listed in the upper left corner along genomic coordinates. Each dot represents single nucleotide polymorphisms and colours indicate linkage disequilibrium (LD; *r²*) with the most likely causal variant, rs12600856, prioritised by multi-trait colocalisation at this locus (darker colours stronger LD). The position of rs12600856 in the genome is highlighted by a red line on the lowest panel. Summary statistics for COVID-19-related outcomes were obtained from the COVID-19 HGI[15], for protein abundances from Pietzner et al.[32], and white blood cell counts from Astle et al.[58]. LD was calculated based on 8350 unrelated white-British participants of the Fenland cohort.

without comorbidities. While no significant effect on the primary endpoint (time to improvement) was detected, patients treated with rhG-CSF experienced significantly fewer severe adverse effects, including respiratory failure, acute respiratory distress symptoms, sepsis, or septic schock[60]. The treatment effect seemed further dependent on baseline lymphocyte counts, with patients <400 per μL benefiting the most. However, leucocytosis was common in the treatment arm, including severe cases. We note, that our results and

the trial are in stark contrast to observational studies associating higher G-CSF plasma levels[78,79] and rhG-CSF treatment among cancer patients with a poor prognosis[80,81], possibly explained by the inability to distinguish cause and effect. We further emphasise, that our genetically anchored drug prioritisation approach cannot make any recommendations about the best time point and dose of intervention during the course of infection/disease, which are crucial parameters for any drug application. Bespoke large randomised clinical trials are warranted to evaluate optimal timing, dosage, and risk-benefit evaluation of rhG-CSF treatment among COVID-19 patients.

Candidate proteins highlighted in the present study might generally act via two distinct mechanisms. Firstly, they may increase/decrease the susceptibility of getting infected with SARS-CoV-2 in the first place, which is also the most powered outcome investigated by the COVID-19 HGI. The effect of BGAT encoded by *ABO* falls most likely into this category[15]. Secondly, once patients are sufficiently infected, host proteins might contribute to exaggerated replication/spreading of the virus into different organ systems or contribute to the hyper-inflammatory response seen in many severe COVID-19 cases with subsequent injury, and possibly failure, of multiple organ systems, including the lung. We observed at least 3-times higher effect estimates of candidate genetic variants for severe COVID-19 compared to testing positive for SARS-CoV-2 for all remaining candidate proteins (Supplemental Data 5), making them likely candidates to contribute to disease severity, which was supported by analysis from the COVID-19 HGI for *ELF5*, *OAS1*, and *SFTP*[15].

Apart from the (refined) annotation of causal genes at known risk loci, establishing a shared signal across different molecular layers and COVID-19 subthreshold findings can reveal yet-to-be-identified risk genes and proteins. For example, we identified *RAB2A*, encoding Ras-related protein Rab2A, as a suggestive causal gene for severe COVID-19, which has only been identified as a genome-wide significant locus while this paper was under review with substantially larger case numbers[82]. While other findings, including *CSF3* (encoding G-CSF), still warrant statistical identification at genome-wide significance for COVID-19 outcomes before being unambiguously declared as genetic risk locus, we argue that establishing convergence of different biological entities at a genetic signal can greatly increase confidence in the plausibility of findings. For example, out of all findings at the *CSF3* locus only the most powered once, that is, white blood cell counts, reach genome-wide significance, although the cluster identified using multi-trait colocalisation aligns with the known biology of G-CSF as a myelopoietic growth factor and was further supported by external trial evidence.

Although the GWAS summary statistics from the COVID-19 HGI represent multiple ancestries and the signal at the *ELF5* locus has recently been replicated in a Brazilian cohort[83], the pQTL instruments are based on a single ancestry and genetic studies of plasma abundances of proteins in other ancestries may reveal additional candidate proteins, that may help to explain the variable prevalence of adverse COVID-19 outcomes across ethnicities[8]. We obtained some evidence that rs766826 might act through a mechanism that is possibly unique to AT2 cells based on an open chromatin region, the concrete underlying mechanism, however, remains elusive. Further studies are needed to decipher the role of rs766826 in the cell-type specific expression of *ELF5*. The same holds true for the suggested mechanisms of action for *ELF5*, for example, co-expression with and possibly regulation of *ACE2* or *TMPRSS2*, that need to be tested in appropriate cellular and animal models, also to investigate the role of *ELF5* in tissues of the respiratory system more in general. Although our results started with the investigation of proteins measured in plasma and might hence provide possible biomarkers for severe COVID-19 in a clinical setting, we did not identify concordant associations based on plasma proteomic profiling for most of the candidates in public data sets[31,84]. This likely reflects the general segregation of proteins that

possibly cause a more severe outcome of COVID-19 than those being a consequence of SARS-CoV-2 infection and COVID-19. We note that while MR can indicate the direction of effects, estimates should be interpreted with caution when plasma/blood is not the tissue of action of the protein or if cis-pQTL(s) can be linked to protein-altering variants or splicing event QTLs[32]. These effects, along with a possible general moderate biological effect, might have contributed to the small effect sizes for BGAT (linked to a splicing QTL) or SFTPD (the cis-pQTL, rs721917, being a missense variant, p.M31T). Finally, while we introduced filters on top of a high PP for a shared signal to ensure robust candidate proteins, correction for multiple testing in statistical colocalisation is still an area of debate and further developments are needed.

Our results demonstrate potential modulators for a poor prognosis among COVID-19 patients with potential therapeutic options. We highlight *ELF5* as a potential regulator in cells that are the primary targets of SARS-CoV-2 by combining population-level genetic evidence with gene expression at single-cell resolution, providing a tangible hypothesis for further functional follow-up studies to investigate the role of *ELF5* for viral entry and wound healing of the epithelial layer of the respiratory system upon severe COVID-19.

## Methods
### Summary statistics for proteins
We obtained locus-specific, ±500kB of the protein-encoding gene, summary statistics from our genome-wide association analysis for 4,775 plasma proteins targeted by the SomaScan v4 assay and 1069 targeted by the Olink proximity extension assay[30,32]. A detailed description can be found elsewhere[30,32]. Briefly, plasma abundances of 4775 protein targets were tested for protein quantitative trait loci using standard GWAS workflows based on 10.4 million single nucleotide polymorphisms (SNPs) among 10,708 individuals of white-British descent. We further obtained genome-wide association statistics for 1069 proteins measured using the complementary Olink technique available among a subcohort of 485 participants. We treated protein assays as separate instances even if those targeted the same protein between both techniques, given the heterogeneity of genetic findings across both platforms[32]. All variants are oriented based on the hg19 genome build.

### Summary statistics for COVID-19
We used four meta-analysed COVID-19 data sets from the June 2021 release of the COVID-19 Host Genetics Initiative including all ancestries but excluding data provided by 23andMe (https://www.covid19hg.org/results/r6/). The phenotypes comprised A2 (Severe COVID-19 vs. population), B1 (hospitalised COVID-19 vs. not hospitalised COVID-19), B2 (hospitalised COVID-19 vs. population), and C2 (SARS-CoV-2 positive vs. population). A summary of case definitions can be found in Supplementary Table 1.

### Statistical colocalisation
To identify genetic variants that are shared between plasma protein levels and the four different COVID-19 definitions, we performed statistical colocalisation[85] in a ±500 kb window around the protein-coding gene as implemented in the R package *coloc* (Fig. 1). We decided for such a 'colocalisation first' to prioritise candidates for formal Mendelian randomisation analysis that fulfil the exchangeability assumption. In other words, to ensure that the effect from the genetic variant to the outcome goes only via the exposure (the protein level) and not via other pathways through proximal or distal variants in linkage disequilibrium, a frequent issue when working with molecular QTLs[86]. In detail, a statistical colocalisation is a Bayesian approach that provides posterior probabilities for each of five hypotheses: H0 – none of two traits has a genetic signal in the region; H1 – only trait 1 has

evidence for a genetic signal in the region; H2 – only trait 2 has evidence for a genetic signal in the region; H3 – both traits have two distinct signals in the same genomic region; and H4 – both traits share the same underlying genetic signal. We used the default prior settings to test for colocalisation ($p_1 = 10^{-4}$, $p_2 = 10^{-4}$, $p_{12} = 10^{-5}$) and tested robustness of findings for candidate proteins by systematically varying across a grid of prior combinations ($p_1 = c(10^{-4}, 10^{-5}, 10^{-6})$; $p_2 = c(10^{-4}, 10^{-5}, 10^{-6})$; $p_{12} = c(10^{-5}, 5 \times 10^{-6}, 10^{-6})$)). To accommodate the single variant assumption of *coloc*, we further required for each protein–outcome pair passing the PP threshold of 80% that respective regional lead variants are in strong LD ($r^2 > 0.8$). While such a filter may drop some true candidates, such as HINT1 for severe COVID-19 (PP H4 = 84%, LD between respective lead signals = 0.79), it filters for likely false-positive findings, such as FAS1 (PP H4 = 93%), that have distinct regional sentinel variants violating statistical colocalisation assumptions ($r^2 = 0.12$).

We optimised computational efficacy by only testing regions with at least suggestive evidence ($p < 10^{-5}$) for either the plasma protein or COVID-19-related outcomes, resulting in a total of 2375 protein targets ($n = 723$ common to both platforms) to be tested. We treated each protein-platform combination as a distinct entity for colocalisation. To avoid spurious colocalisation results, we further ensured that the lead cis-pQTL or a proxy in strong LD ($r^2 > 0.8$) was included in the overlapping set of SNPs used for colocalization with COVID-19 summary statistics. We took only that protein–outcome regions forward with sufficient presentation of the lead cis-pQTL signal ($r^2 > 0.8$) and further provide similar information for COVID-19 statistics. We further kept only SNPs in the COVID-19 summary statistics present in 80% of the contributing studies. This resulted in a total of 1121 unique protein targets with reliable colocalisation results for at least one of the four COVID-19 outcomes. We note, that due to the varying sets of overlapping SNPs between protein and COVID-19 summary statistics, we chose to take forward the SNP most strongly associated with plasma protein levels as a representative to present effect estimates. While this may have led to slightly varying candidate SNPs for each protein–outcome pair, the fact that we rigorously filtered for a preserved protein signal ensures that all those variants should tag the same underlying potentially causal variant. We used the largest Fenland subset ($n = 8350$) to compute LD information, if SNPs were not available, we queried the non-Finnish European sample from the 1000 Genomes project as implemented in the R package *ieugwasr*. We report minor allele frequencies based on the Fenland data set.

We finally categorised candidate proteins (i.e., PP H4 > 80% and LD regional sentinels $r^2 > 0.8$) into three tiers based on statistical significance: 1 – genome-wide significant ($p < 5 \times 10^{-8}$) in protein and COVID-19 summary statistics; 2 – genome-wide significant in either protein or COVID-19 statistics; and 3 – suggestive candidates with subthreshold ($5 \times 10^{-8} < p < 10^{-4}$) findings for both.

## Multi-trait colocalisation

We used hypothesis prioritisation in multi-trait colocalisation (HyPrColoc)[33] at selected protein loci (1) to identify a shared genetic signal across various traits, including gene expression, plasma protein levels, COVID-19 outcomes, and other phenotypes, and (2) to test whether phenotypes with genetic signals at the same genetic locus centre around distinct causal variants, that is, although in close proximity represent distinct genetic findings. HyPrColoc provides for each cluster three different types of output: (1) a posterior probability (PP) that all phenotypes in the cluster share a common genetic signal, (2) a regional association probability, that it, that all the phenotypes share an association with one or more variants in the region, and (3) the proportion of the PP explained by the candidate variant. We considered a highly likely alignment of a genetic signal across various phenotypes if the PP > 80% and report obtained PPs otherwise. We

further used the intrinsic fine-mapping approach done by HyPrColoc to report candidate causal variants at each locus.

## Mendelian randomisation

To derive effect directions and estimate possible effects of lifelong higher/lower protein abundances on COVID-19 susceptibility and severity, we performed single-instrument MR analysis using cis protein quantitative trait loci (cis-pQTLs) as instruments. We computed the ratio between the effect of the genetic instrument on the outcome divided by the effect on the exposure (so-called Wald ratio[87]) to derive an estimate for the causal effect of a 1 s.d. increase in plasma abundances of the candidate protein on the risk for COVID-19.

## Tissue gene expression

We incorporated gene expression data by testing for a shared genetic signal between protein abundance in plasma and expression of the protein-encoding gene in one of at least 49 tissues of the GTX v8 resource[35]. We used the same colocalisation approach as described above.

## Multi-ethnic fine-mapping of the ELF5 locus

We used Bcftools v1.9 to isolate variants present in the *ELF5* locus (chr11:34440000-34540000) from the 1000 G project GRCh37 phase 3 official release data, stratifying for European and African ancestry[88,89], and isolated biallelic variants using Plink v2.00a22.3LM. To calculate the linkage disequilibrium ($r^2$) for all combinations of variants we processed all variants with Tomahawk v0.7.1 (https://mklarqvist.github.io/tomahawk/). We obtained missing linkage disequilibrium statistics using the R package *LDLinkR*.

We used FINEMAP v1.4[90] to perform multi-ancestry fine-mapping following a previously published workflow[25]. Summary statistics and linkage disequilibrium statistics by ancestry were organised into files formatted for use. We started with the most powered European ancestry data, setting prior probabilities for each variant to be equal. We used the PIPs from this analysis as priors in the subsequent analysis of the African ancestry results. We note that the added value of trans-ethnic fine-mapping is likely due to smaller LD-blocks in participants of African descent, since we identified five SNPs co-segregating with rs766826 in Europeans ($r^2 > 0.7$), while none did so using the African reference panel.

## Phenotypic follow-up of candidate cis-pQTLs

We systematically tested for phenotypic associations for cis-pQTLs by querying the OpenGWAS[38] database, including proxies in high LD ($r^2 > 0.8$). To test for a shared genetic signal between the FEV1/FCV ratio (a proxy for lung function) and plasma levels of ELF5, we downloaded genome-wide summary statistics from Shrine et al.[39]. We conditioned on two stronger independent lead signals in the region (rs10836366 and rs1648123) to account for the single variant assumption in statistical colocalisation.

## Collation of target genes of ELF5

We collated a list of genes with possible direction association with *ELF5* by querying the Molecular Signatures Data Base[91], the Enricr tool[92], the Harmonizome[93], including ChIP-Seq experiments[94], and a curated gene co-expression network[95] (Supplementary Data 7).

## Single-cell/nucleus RNA sequencing quality control

Single-cell/nucleus RNA sequencing (sc/nRNA-seq) healthy control data sets of the olfactory mucosa (GSE139522), nasopharynx (EGAS00001005461), and lungs (EGAS00001004689; EGAS00001004419: SAMEA6848756, SAMEA6848761, SAMEA6848765, SAMEA6848766) were reanalysed in this study[41–44]. Due to the increased noise of snRNA-seq data, we performed ambient RNA removal on the lung data set with SoupX v1.4.5[96]. Analysis was performed with Seurat

v3.1.4[97,98]. For the olfactory mucosa and lung data sets, individual samples were integrated and annotated from scratch. Individual samples were subjected to an upper bound filter of <10% mitochondrial reads and >200 genes expressed, and an upper bound filter of 3000–6000 genes depending on the sample.

### Single-cell/nucleus RNA sequencing analysis

After log-normalisation and scaling, canonical correlation analysis was used for integration and batch-correction of the individual samples. Principal component analysis and Uniform Manifold Approximation and Projection for dimension reduction (UMAP) were calculated for each integrated data set. Finally, after unsupervised clustering, the cell-type assignment was performed as previously described[16,41–44,61] and marker genes are depicted in Supplementary Fig. 13. Differentially expressed genes were identified using a MAST-based differential expression test. The Pearson's correlation was calculated on log-normalised expression values for all detected genes against *ELF5* in Sustentacular cells, which expressed *ELF5* the highest. Gene set enrichment analysis[91] (GSEA; v4.1.0) was used to test for enrichment of the collated *ELF5* target genes against all detected genes where the weights used were the Pearson's correlation values. Utilising the collated *ELF5* target genes, cell-type specific pathway enrichment was performed using Metascape[99].

### Immunohistochemistry

Postmortem olfactory mucosa and lung tissue were collected from control and COVID-19 donors (Supplementary Table 2). All donors or their next of kin/legal representative gave consent. COVID-19 status from the controls was assessed by Spindiag Rhonda PCR rapid COVID-19 test according to the manufacturer's protocol. This study was approved by the local ethics committees (EA1/144/13, EA2/066/20 and EA1/075/19) as well as by the Charité–BIH COVID-19 research board and is in compliance with the Declaration of Helsinki; autopsies were performed on the legal basis of §1 of the Autopsy Act of the state Berlin and §25(4) of the German Infection Protection. Control lung tissues were purchased from OriGene (TissueFocus) and Tissue Solutions. Samples were embedded in paraffin and sectioned at a 5µm thickness. Sections were deparaffinized in Roticlear (CarlRoth, A538.1) and rehydrated with an ethanol series. Antigen retrieval was performed by submerging slides in 10 mM Sodium Citrate Buffer (Sigma-Aldrich, C999-1000ML) at 95 °C for 10 minutes and left to cool for 30 minutes. Sections were permeabilized and blocked with 5% goat serum PBST (0.5% Triton X-100 in PBS) for 1 hour. Primary antibodies (1% goat serum PBST) were then added to the tissues and left to incubate overnight. This was followed by secondary antibody (1% goat serum PBST) labelling for 1 hour at room temperature. Antibodies used in this study and their respective dilutions can be found in Supplementary Table 3. TMPRSS2 with ACE2 staining was performed according to the instructions of VectaFluor™ Excel Amplified Kit (Vector Laboratories; DK-2594). To remove autofluorescence, sections were sequentially treated with Lipofuscin Autofluorescence Quencher (PromoCell, PK-CA707-23007) and Vector TrueVIEW™ Kit (Vector Laboratories, SP-8400). Sections were then stained with 16 µM Hoechst 33258 (ThermoFischer, H3569) for 5 minutes, washed with 1× PBS, and mounted with VECTASHIELD® HardSet™ Antifade Mounting Medium (Vector Laboratories, H-1400-10). Stained slides were visualised with a Leica SP8 confocal microscope. Images were processed and assembled with FIJI v1.0[100].

### Reporting summary

Further information on research design is available in the Nature Research Reporting Summary linked to this article.

## Data availability

Summary statistics for protein levels are available from https://omicscience.org/apps/pgwas/ (SomaScan v4) and https://zenodo.org/record/6787142#.Yr761uxBxhE (Olink). Summary statistics for COVID-19 are available from https://www.covid19hg.org/results/r6/. scRNAseq data sets are available under the accession IDs: Olfactory Mucosa (GSE139522); Nasopharynx (EGAS00001005461); Lung (EGAS00001004689) and (EGAS00001004419). Publicly available GWAS summary statistics were obtained from the IEU OpenGWAS project (https://gwas.mrcieu.ac.uk/).

## Code availability

Associated code and scripts for the analysis are available on GitHub (https://github.com/pietznerm/elf5_covid_19)[101].

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

## Acknowledgements

We are grateful to all Fenland volunteers and to the General Practitioners and practice staff for assistance with recruitment. We thank the Fenland Study Investigators, Fenland Study Co-ordination team and the Epidemiology Field, Data and Laboratory teams. Proteomic measurements were supported and governed by a collaboration agreement between the University of Cambridge and SomaLogic. The Fenland Study (10.22025/2017.10.101.00001) is funded by the Medical Research Council (MC_UU_12015/1). We further acknowledge support for genomics from the Medical Research Council (MC_PC_13046). CL, EW, MP, and NJW are funded by the Medical Research Council (MC_UU_00006/1 - Aetiology and Mechanisms). Non-COVID-19 autopsy samples were provided by the BrainBank/BioBank of the Department of Neuropathology at the Charité – Universitätsmedizin Berlin, funded by the Deutsche Forschungsgemeinschaft (DFG, German Research Foundation) under Germany´s Excellence Strategy – EXC-2049 – 390688087. COVID-19 autopsies were supported by the German Network University Medicine

NUM FKZ 01KX2021 Organostrat and Defeat Pandemics (H.R., M.A.M., and F.L.H.) and by the Federal Ministry of Education and Research within the framework of the network of university medicine (DEFEAT PAN-DEMICs, 01KX2021 and NATON, 01KX2121). This work was supported by the German Ministry for Education and Research through the Medical Informatics Initiative (junior research group "Medical Omics" to S.L., 01ZZ2001). The Richards research group is supported by the Canadian Institutes of Health Research (CIHR: 365825; 409511, 100558, 169303), the McGill Interdisciplinary Initiative in Infection and Immunity (MI4), the Lady Davis Institute of the Jewish General Hospital, the Jewish General Hospital Foundation, the Canadian Foundation for Innovation, the NIH Foundation, Cancer Research UK, Genome Québec, the Public Health Agency of Canada, McGill University, Cancer Research UK [grant number C18281/A29019] and the Fonds de Recherche Québec Santé (FRQS). JBR is supported by a FRQS Mérite Clinical Research Scholarship. Support from Calcul Québec and Compute Canada is acknowledged. TwinsUK is funded by the Wellcome Trust, Medical Research Council, European Union, the National Institute for Health Research (NIHR)-funded BioResource, Clinical Research Facility and Biomedical Research Centre based at Guy's and St Thomas' NHS Foundation Trust in partnership with King's College London. These funding agencies had no role in the design, implementation or interpretation of this study. J.D.S.W. is a graduate student in McGill University's Quantitative Life Sciences program, whose work is supported by a CIHR grant.

## Author contributions

Conceptualisation: C.L., M.P., C.C. Data curation/software: M.P., R.L.C., S.L., E.W. Formal analysis: M.P., R.L.C., S.L., E.W., J.D.S.W. Methodology: S.L., S.T. Visualisation: M.P., R.L.C., S.L., K.J. Experiments: R.L.C., K.J., H.R. Funding acquisition: C.L., N.J.W., C.C., I.L., R.E. Project administration: C.L., N.J.W., C.C., R.E., I.L., F.L.H. Supervision: C.L., C.C. Writing – original draft: M.P., R.L.C., S.L., C.C., C.L. Writing—review & editing: E.W., H.R., S.T., B.H., H.Z., R.E., M.M., J.B.R., F.L.H., L.S., I.L., N.J.W.

## Funding

## Competing interests

E.W. is now an employee of AstraZeneca. J.B.R.'s institution has received investigator-initiated grant funding from Eli Lilly, GlaxoSmithKline and Biogen for projects unrelated to this research. He is the CEO of 5 Prime Sciences (www.5primesciences.com), which provides research services for biotech, pharma and venture capital companies for projects unrelated to this research. The remaining authors declare no competing interests.
