## [Peer Review File · Nature Communications]

ELF5 is a potential respiratory epithelial cell-specific risk gene for severe COVID-19REVIEWER COMMENTS

Reviewer #1 (Remarks to the Author):

Pietzner and colleagues performed a statistical colocalization analysis between two GWAS summary statistics for plasma protein levels (one of which was previously published by the authors using data from UK Biobank) and four GWAS summary statistics recently made publicly available by the COVID-19 Host Genetics Initiative (HGI) via a website and pre-print publication (<https://www.medrxiv.org/content/10.1101/2021.11.08.21265944v1>; <https://www.covid19hg.org/results/r6/>). This was followed by a Mendelian randomization analysis across the summary statistics and a reanalysis of publicly available single-cell/nuclear RNA sequencing data from EGA and other sources (COVID-19 Cell Atlas?). To confirm the presence of ELF5 protein, immunohistochemical staining experiments were performed on postmortem olfactory mucosa and lung tissue from four controls and four COVID-19 donors.

Although I initially read the abstract with enthusiasm, I am a bit disappointed with the actual innovations of this study, which only becomes clear when reading the whole manuscript. The abstract sounds too grandiose, as the "large-scale human genetic studies" are the GWAS summary statistics already published by the HGI.

The association of the ELF5 gene and its lead SNP rs766826 (intronic variant of ELF5) reported here as new findings are not new findings and have been reported as genome-wide significant associations in the HGI preprint and on the HGI website (<https://app.covid19hg.org/variants>), which was not mentioned here. Variant rs766826 has been further reported as an eQTL variant for ELF5 (<https://www.medrxiv.org/content/10.1101/2021.11.08.21265944v1>; Supplementary Table 5), which should be mentioned here.

I have the following comments:

Although the authors present many interesting results, I found the manuscript difficult to follow in some parts and also felt that the summary and discussion did not clearly state which data and which results were truly new and that mainly published summary statistics and published single-cell data were reanalyzed together.

The ELF5 gene and the lead SNP rs766826 studied here were recently identified by the COVID-19 Host Genetics Initiative (HGI) as a genome-wide significant risk locus/lead SNP for COVID-19 severity (hospitalized COVID19+ compared with population controls; preprint available at <https://www.medrxiv.org/content/10.1101/2021.11.08.21265944v1>; see ELF5/rs766826 in Figure 2 of the preprint; ELF5/rs766826 are listed together under new associations at <https://app.covid19hg.org/variants>). This should be clearly mentioned in the abstract and discussion section, as the exact same HGI GWAS summary statistics were used in this study.

It should also be mentioned that the lead variant rs766826 was found to be an eQTL variant for ELF5 already in the study of the HGI (<https://www.medrxiv.org/content/10.1101/2021.11.08.21265944v1>; Supplementary Table 5).

Line 45: "establishing a shared genetic architecture at protein-coding loci using large-scale human genetic studies" This does not indicate that colocalization analysis of GWAS summary statistics (with 500kb regions around genes; including non-protein-coding variants) was performed. Neither rs766826 nor rs7947771 are protein-coding variants, which also leads to misunderstanding.

Line 48: The ">4-fold higher risk (odds ratio: 4.85 ..." refers to a variant named rs7947771 from

Suppl Table 2, but variant rs766826 is always discussed as the lead cis-pQTL, leading to misunderstanding.

Line 57: Sometimes "COVID-19 prognosis" is used, sometimes " COVID-19 severity" is used throughout the text.

Line 112: Supplementary Table 1 lacks the final colocalisation results showing that the 8 proteins mentioned are the top 8 results from colocalisation screenings. A supplemental table with all full 2375 proteins times 4 tested GWAS HGI sumstats results should be shown to demonstrate that only 8 proteins mentioned here had suggestive evidence ($p < 10^{-5}$).

Line 123: odds ratio: 4.85; 95%-CI: 2.65-8.89; p -value $< 3.1 \times 10^{-7}$ refers to variant rs7947771 (Suppl Table 2; also in the Abstract) but not to the discussed ELF5 variant rs766826 (rs766826 shown in Figure 3 and Discussion). I find it confusing when statistical effects are given sometimes for this variant and sometimes for that variant.

Line 429: Number of proteins tested is sometimes 2,375 proteins, sometimes 2,345 as in Figure 2. Please correct.

Line 446: cis-pQTL colocalization p value threshold ($p < 10^{-5}$) needs to be further divided by 4 because 4 HGI GWAS sumstats were used for colocalization screening.

Line 131: Please indicate in Figure 2 which of the two original GWAS protein level sum statistics was used to find the result and whether it was significant in both studies. Please indicate in Figure 2 which P value is shown (from MR?) and which probability is meant here (posterior probability H3 or H4 from colocalization analysis?).

Line 162: What does SNP rs35725681 in Figure 3 mean? The first 3 GWAS sum statistics shown in Figure 3 are exactly taken from the published HGI GWAS sum statistics, which should be mentioned in the legend.

Line 142: „The lead cis-pQTL for ELF5, rs766826 (MAF=35.9%), is in strong linkage disequilibrium (LD; $r^2=0.81$) with a recently identified variant rs61882275 associated with severe COVID-19 in an independent study using whole genome sequencing³². The causal gene remained unidentified, ...“ Reading this, I have the impression that rs766826 and ELF5 are novel findings reported here for the first time and refine a genetic signal from a bioRxiv study of doi:10.1101/2021.09.02.21262965. However, rs766826 and ELF5 have been previously reported in the HGI GWAS study from which the GWAS summary statistics are taken here (see above). Please cite HGI bioRxiv.

Line 184: „We observed that ELF5 was almost exclusively expressed by different epithelial cells of the respiratory system“ is true only to these selected single cell datasets presented here (from the COVID-19 Atlas?). ELF5 is also expressed in breast tissue, bladder, skin and various other tissues, see <https://gtexportal.org/home/gene/ELF5>.

Line 201: Figure 4 shows previously published data, not data from new experiments. Please indicate in the legend.

Line 311: „We identified six not previously reported and replicated two known causal genes and“. Not correct, please cite HGI COVID-19 release 6.

Line 316: „We refine the association of genetic variation at ELF5 with severe COVID-19 to a single causal variant (rs766826) with high confidence (PP=99%)“. These PP results also apply to SNP rs766826 already as eQTL from the HIG release6 bioRxiv Paper, Supplementary Table 5, line 56 and

57

(<https://www.medrxiv.org/content/medrxiv/early/2021/11/11/2021.11.08.21265944/DC2/embed/media-2.xlsx?download=true>) and are therefore not new findings.

Line 335: "our scnRNAseq data provide strong evidence that ELF5 is expressed ...". Please reword as these single cell RNAseq data are not new and have already been published by others.

Line 461: "49 tissues of the 461 GTX v8 resource". The authors do not show that actual completely different tissues show expression for ELF5, why here the conclusion that ELF5 only for respiratory system, such as secretory and alveolar type 2 cells, when single cell data were studied only from these cells?

<https://gtexportal.org/home/gene/ELF5>

Line 742: Links to the published datasets cannot be found clearly. The GitHub page does not exist. Please provide exact links.

Line 742: The links to the published datasets do not point to the records. Also the GitHub page does not exist. Please provide correct links.

Reviewer #2 (Remarks to the Author):

The authors report eight candidate proteins that are associated with COVID-19 outcomes. The authors leverage data on genetic variants and their association with protein levels in plasma, as this can be used to predict exposure of individuals to lifelong differences in protein levels in the tissue. The authors highlight a candidate protein ELF5 that has evidence for positive causal association with critical illness due to COVID-19, supported by further single cell RNA sequencing analysis and immunohistochemistry assays in lung showing specific expression in relevant cell types, and colocalization/co-expression with host viral entry proteins in cells susceptible to infection.

These approaches are important because they aim at disentangling the causal mediators of disease from the plethora of association results produced by large scale genome-wide association studies (GWAS) of host genetics of COVID-19. Better understanding of causal mechanisms behind infection and progression to severe disease can inform repurposing existing drugs and, in the long run, the creation of new ones that could benefit patients and prevent critical illness due to the virus, and finding biomarkers of disease progression. The analytical framework of the study is appropriate and the authors have undertaken a significant effort towards disentangling some of the host factors for COVID-19. I have questions regarding the methods, and some suggestions that I feel could improve the discussion of findings.

1) I think it would be best to list out all candidate proteins out in the beginning of the results section within the main text. Including them here makes it easier for the reader to immediately know and refer back to these proteins when reading through the results/discussion instead of searching for them in the figures.

2) The authors use data for four COVID-19 outcomes released by the COVID-19 Host Genetics Initiative (HGI) that encompasses the largest sample sizes for any COVID-19 host genetic GWAS at this time. These four summary statistics were produced from analyses with highly overlapping samples (A2 included in B2 and B2 included in C2). Whilst running the analyses of this manuscript on all available summary statistics can be valuable in detail, the comparison of B1 analysis to the other outcomes is tricky due to the different definition of controls. Deeper discussion of these differences in ascertainment and what their implications could mean for the presented results would be needed. My suggestion would also be to keep B1 results in the supplementary tables but for ease of comparison

across the other three case-definitions it would be clearer to show only analyses with the same definition of controls in the main text and Figure 2.

3) Figure 1 has mislabelled the A2 (shown as A1) and C2 (shown as C1), but I actually suggest removing them from the main text and figures as they are only meaningful to readers who know about the HGI analysis namings. These are fine in the methods and supplement. The naming of the outcomes should also be carefully checked throughout the manuscript for consistency as it is not always clear which outcome the authors refer to, e.g. HGI 'critical illness' (A2) definition seems to appear as 'severe COVID-19' and 'more severe COVID-19 prognosis'. I would suggest keeping to the original definitions when talking about the outcomes.

4) It would be valuable to discuss in the manuscript how the COVID-19 outcome definitions used may differ from the actual COVID-19 phenotype that the genetic locus is affecting; deciphering whether a genetic locus found in the GWAS for a particular COVID-19 outcome is associated with susceptibility to viral entry and host infection or to the progression to severe disease after infection is difficult due to the nature of infectious disease and the phenotypes that can be collected. In addition, due to the overlap in samples between the HGI analyses (and ascertainment of samples, different overall sample sizes, etc), comparing the four COVID-19 GWAS summary statistics should be accompanied with discussion on how these can affect the interpretation. For example, a locus contributing to differences in susceptibility to infection may be also present in critically ill vs. population outcome analysis, and a locus contributing to progression to more severe disease upon infection may be present in all cases vs. population outcome because of the inclusion of severe cases.

There have been some efforts to address this, e.g. the HGI has conducted in their update manuscript a Bayesian analysis to decipher which of the genetic loci is affecting infection susceptibility and which are associated with progression to severe disease. For example, in that analysis *ELF5* locus is associated with disease severity and the *ABO* locus with infection susceptibility. This reviewer thinks that incorporating such information from published literature into the narrative of the manuscript could aid the interpretation of key figures (e.g. Figure 2) and conclusions of the article, e.g. do the MR findings support what has been previously reported? I think it is worthwhile to at least distinguish the measured outcomes from the underlying implication. E.g. in the introduction lines 97-98 when discussing outcomes the range is 'all positive cases' to confirmed COVID-19 positive and hospitalized with respiratory support or death', whilst the locus-specific effects might be 'increasing susceptibility to infection' or the 'risk of progression to severe disease once infected'.

5) Partially overlapping with the previous point, how much do the authors think the differences in GWAS sample sizes, overlapping samples between outcomes, coverage at each locus (how many studies out of all participating studies contributed data at that single instrument) can affect colocalization and MR in addition to the ancestry differences that the authors mention?

6) The authors discuss the ancestry differences between HGI summary statistics and the cis-pQTL data. Have the authors considered using the EUR-only summary statistics from HGI (I believe these are available for some of the population controls analyses)?

7) How were the GWAS regions for coloc analysis defined? It was not mentioned in the methods, whether e.g. any P-value threshold was used to define regions around GWAS peaks. For example the G-CSF pQTL is colocalizing with a GWAS region that does not reach genome-wide significance. Could the authors comment on whether they have any concerns over the interpretation of coloc/MR results from GWAS loci that do not approach significance thresholds?

8) Did you consider any multiple testing correction for MR analysis? What did you consider as the threshold for significance?

9) Figure 2 shows the results of the MR analysis of association of predicted protein levels with risk of COVID-19 outcomes.

a) Current ordering is based on Gene/protein name alphabetical order. I would suggest ordering based on biology e.g. the above mentioned infection susceptibility and severity locus information or the statistical association strength of the COVID-19 loci, or something else. I suggest this because it could help see patterns in both panels, especially the right hand side one displaying shared/distinct signal.

b) The term 'shared genetic architecture' is often used when discussing results of genetic correlation analyses, and therefore it would be important to add a note to the legend and/or methods and main text that you are referring to colocalization probability PP.H4 as shared signal and PP.H3 as distinct signal.

c) Please report in the legend that the P-value is referring to MR analysis and the type of test so that this is easier to find in the methods.

d) The analysis results are plotted for each protein in reverse order (most severe COVID-19 outcome last) to how the results are displayed in Supplementary Table 2 (most severe outcome first). It would be easier to cross reference these if they were reported in the same order.

10) Please define on line 158 what you mean by 'unrelated', i.e. was a P-value for a particular test not significant?

11) Authors refer to putative causal variants in the manuscript but it is unclear which method was used to define causal variants (i.e. was separate fine-mapping performed or was this run as part of coloc?). If this was given directly as the colocalization output, it would be useful to mention this in the methods and report the posterior probabilities for each SNP in Supplementary Table 2 (SNP.PP.H4). Also at times different variants are referred to as candidate causal variant at a locus, e.g. at the G-CSF locus the Supplementary Figure 1 refers to rs4795412 whereas Supplementary Table 2 contains variants rs3826331 and rs8081692. What is behind the difference?

12) The authors' findings and previous literature points to AT2 cells as being important for severe COVID-19 disease outcomes. It would be helpful to orient the reader to lung biology by adding a phrase or two about the role of these cells in healthy tissue (or also in disease) when these cells are first described in the main text.

13) Figure 4:

a) Panel C: Can ACE expression be shown as a gradient similarly as ELF5 in panel B?

b) Panel D: Could you also add to the legend or methods a note how the genes on the right hand side panel (TMPRSS2, FURIN, NRP1, NRP2) were chosen for display as this is not obvious with the exception of the TMPRSS2?

14) The authors highlight an intriguing possible explanation for the tissue specificity of ELF5 expression. It would be great to outline the crucial region more clearly in Supplementary Figure 4, as it is a bit difficult to interpret without more direction. Should the reader consider the lung open chromatin regions as a whole (all highlighted area) or the block just to the right of the SNP of interest? The tissues could also be ordered on the y-axis by expression level in the tissue from highest to lowest to make the comparison between lung and other highly expressing tissues mentioned in the main text easier.

15) Regarding Figure 6 and for phenome-wide colocalization analysis at CSF3 locus:

a) I'm wondering whether it would be more informative to show the non-standardized z-scores, so as to get an impression of the relative effects of variants in this locus on the different phenotypes? Currently from what I understand the figure mainly shows that variants in higher LD with the candidate causal variant also have stronger association with the trait as is expected based on the definition of LD. For that reason also most of the molecular traits in the figure have a similar

LD/zscore pattern as they presumably come from the same study.

b) Additionally for the COVID-19 traits, the z-scores seem to distribute differently in the group of variants in high LD to the candidate variant. Can this be explained by mixed ancestry included in the meta-analysis?

c) Methods should include how LD was estimated here.

d) Was any multiple testing correction applied to phenome wide colocalization analysis?

e) It would make sense to include in brackets in the main text that the G-CSF is in CSF3 locus to better link this figure to the main text (as currently one has to look at Figure 1 to figure this out).

f) The legend title says "GCSF3" - is this a typo? Also "gene variant" is probably a typo and to be replaced with "genetic variant" as it is not in a gene.

g) The ordering of the traits on the figure should be grouped more logically e.g. all neutrophil phenotypes together (or even better, select a few representative ones).

16) Supplementary Table 4 does not include the COVID-19 outcomes plotted in Figure 6, so these should be added.

17) Something that I found confusing at first was that for all stacked colocalization plots, it seems that sometimes the lighter filled points (lower r^2) can look very dark because they are clustered together and the dark borders make them look like a dark point. It made the figure look like some high r^2 variants have low P-values. Would it be possible to modify the border colour or alternatively narrow in on the locus to spread the points out more to make sure the light colour comes out?

18) Supplementary Figure 3: Should be noted in the legend that the bar represents where this other variant rs35725681 is. It could be also worth highlighting the likely causal variant for ELF5 pQTL to give more context to this region. Maybe highlight both variants with diamonds to show which variant it is instead of the bar?

19) Lines 175-182: Curiously, a similar counterintuitive effect has been seen for IPF associated MUC5B locus by Fadista et al (<https://doi.org/10.1016/j.ebiom.2021.103277>).

20) Typo on line 486? Seems like it should be referring to Supplementary Figure 12.

21) Discussion:

a) Repetition of content on lines 311-315 and 324-327 regarding results, and on lines 322 and 331 about Ets transcription factor family.

b) On lines 393-397: Could part of the interpretation of MR effect estimate being low also be that the GWAS effect points to infection susceptibility (instead of disease progression) is on its own relatively modest?

c) Line 334: Statement about embryonic lethal mouse is missing a reference.

d) Lines 342-345: Is this missing a reference to mouse study? Can you specify how the findings partly align as this phrase is a bit difficult to interpret.

e) Line 350: Unclear what "if any" is referring to.

f) Line 352: Likely authors mean "infection symptoms" as "infectious symptoms" implies anosmia is infectious. Also, what is the symptom highly specific for?

g) Lines 366-369: The wording here is a bit speculative as the authors say ELF5 is likely one of many contributors but also only possibly contributing to anosmia, so I recommend rephrasing.

h) Lines 409-410 again referring back to my comments about susceptibility to infection/progression to severe disease, I would add specifically the proteins here that have clearer evidence of modulating disease severity and not infection susceptibility if this is the desired message, or mention that some loci have more evidence of being associated with infection susceptibility.

i) ELF5 has been highlighted as a candidate gene in the region before by two GWAS mentioned in the text, and HGI paper which mentions ELF5 fills the criteria "closest gene to the lead SNP, and an eGene i.e. fine-mapped cis-eQTL variant PIP > 0.1 in GTEx Lung, in LD with a COVID-19 lead variant r^2 >

0.6)". Whilst this study is the first to follow up with pQTL and RNA analyses and staining, this reviewer is leaning towards thinking the protein candidate is perhaps not 'novel' but the current study significantly strengthens the evidence for this gene product.

Methods

22) Please describe in the methods you are using COVID-19 summary statistics versions with all ancestries and 23andMe excluded. Please also include in the methods and supplementary table legends that you are using hg19 genome build versions and the same for cis-pQTL.

23) A description of the method and analysis for "multi-trait colocalization" analysis mentioned on lines 126 and 293 and referred to as Reference number 31 seems to be missing from the methods section.

24) I would recommend breaking down the "Single-cell/nucleus RNA sequencing" section into two separate parts covering the quality control and then analyses. Also good to refer in the methods to the same analysis name as in the main text, e.g. main text talks about cell-type specific gene set enrichment analysis, whilst the methods talks about gene set enrichment analysis. This would make it easier for the reader to confidently find the right section in the methods.

References:

25) References #14 and #30 to be fixed, the author is the consortium 'COVID-19 Host Genetics Initiative'

26) Lines 78-79 of the main text: I would add the Zhou and Downes paper references where the genes are mentioned in this sentence to help the reader identify which study points to which gene. Authors could also consider Kasela et al PMID: 34425859 when discussing chr3 locus and SLC6A20

REVIEWER COMMENTS

Reviewer #1 (Remarks to the Author):

Pietzner and colleagues performed a statistical colocalization analysis between two GWAS summary statistics for plasma protein levels (one of which was previously published by the authors using data from UK Biobank) and four GWAS summary statistics recently made publicly available by the COVID-19 Host Genetics Initiative (HGI) via a website and pre-print publication (<https://www.medrxiv.org/content/10.1101/2021.11.08.21265944v1>; <https://www.covid19hg.org/results/r6/>). This was followed by a Mendelian randomization analysis across the summary statistics and a reanalysis of publicly available single-cell/nuclear RNA sequencing data from EGA and other sources (COVID-19 Cell Atlas?). To confirm the presence of ELF5 protein, immunohistochemical staining experiments were performed on postmortem olfactory mucosa and lung tissue from four controls and four COVID-19 donors.

Although I initially read the abstract with enthusiasm, I am a bit disappointed with the actual innovations of this study, which only becomes clear when reading the whole manuscript. The abstract sounds too grandiose, as the "large-scale human genetic studies" are the GWAS summary statistics already published by the HGI.

The association of the ELF5 gene and its lead SNP rs766826 (intronic variant of ELF5) reported here as new findings are not new findings and have been reported as genome-wide significant associations in the HGI preprint and on the HGI website (<https://app.covid19hg.org/variants>), which was not mentioned here. Variant rs766826 has been further reported as an eQTL variant for ELF5 (<https://www.medrxiv.org/content/10.1101/2021.11.08.21265944v1>; Supplementary Table 5), which should be mentioned here.

R1 response 1 *We of course agree with the reviewer that we are not the first to identify the ELF5 locus to be associated with COVID-19 and have now thoroughly rephrased any section in the manuscript that could have given the impression that we did so, paying appropriate tribute to the work of the COVID-19 HGI and other efforts who reported this locus. We have further received positive feedback from the lead authors of the flagship paper that there are no concerns about us reporting on ELF5 (email can be provided).*

We now make it more clear that the contribution of our study lies in addressing major bottlenecks of GWAS, as also pointed out by reviewer #2: 1) the identification of candidate causal genes via integration of cis-pQTLs at reported or unreported risk loci for COVID-19, 2) the bespoke follow-up of the strongest candidate, ELF5, identifying not only the likely affected tissue, but the specific single cell population of the respiratory system, including divergent morphological changes associated with ELF5 expressing cells in deceased COVID-19 patients, and 3) provide translational insights by prioritizing human recombinant G-CSF as a promising treatment option in line with a recent clinical trial.

I have the following comments:

Although the authors present many interesting results, I found the manuscript difficult to follow in some parts and also felt that the summary and discussion did not clearly state which data and which results were truly new and that mainly published summary statistics and published single-cell data were reanalyzed together.

R1 response 2 *In response to the referees' comment, we have thoroughly revised the manuscript to clarify the narrative and present findings accordingly. We made it further clear what type of data has been used and whether this was based on published resources or newly generated, such as all immunostainings. We note that only the unique combination of all three resources, that is, COVID-19 GWAS, protein GWAS, and scRNAseq data, along with long-standing domain expertise, enabled the discoveries made in this study and none of the datasets, published or not, would have been sufficient on its own, to understand a/the possible role of ELF5 and other proteins in COVID-19.*

The ELF5 gene and the lead SNP rs766826 studied here were recently identified by the COVID-19 Host Genetics Initiative (HGI) as a genome-wide significant risk locus/lead SNP for COVID-19 severity (hospitalized COVID19+ compared with population controls; preprint available at <https://www.medrxiv.org/content/10.1101/2021.11.08.21265944v1>; see ELF5/rs766826 in Figure 2 of the preprint; ELF5/rs766826 are listed together under new associations at <https://app.covid19hg.org/variants>). This should be clearly mentioned in the abstract and discussion section, as the exact same HGI GWAS summary statistics were used in this study.

R1 response 3 *As outlined in our comment above, we thoroughly revised the paper to pay appropriate tribute to the results provided by the COVID-19 HGI (page 2, lines 45-47; page 5 line 116 and lines 130-135).*

It should also be mentioned that the lead variant rs766826 was found to be an eQTL variant for ELF5 already in the study of the HGI (<https://www.medrxiv.org/content/10.1101/2021.11.08.21265944v1>; Supplementary Table 5).

R1 response 4 *We now refer more clearly to the eQTL assignment reported by the COVID-19 HGI based on data from the GTEx consortium (page 5, lines 1-33-134).*

Line 45: “establishing a shared genetic architecture at protein-coding loci using large-scale human genetic studies” This does not indicate that colocalization analysis of GWAS summary statistics (with 500kb regions around genes; including non-protein-coding variants) was performed. Neither rs766826 nor rs7947771 are protein-coding variants, which also leads to misunderstanding.

R1 response 5 *We acknowledge that our previous wording was imprecise and have rephrased the corresponding section accordingly to improve clarity (page 2, lines 45-47).*

Line 48: The ">4-fold higher risk (odds ratio: 4.85 ..." refers to a variant named rs7947771 from Suppl Table 2, but variant rs766826 is always discussed as the lead cis-pQTL, leading to misunderstanding.

R1 response 6 We apologize for the inconsistency, which was a legacy of the computational pipeline used to run cis-based colocalization. We now report at each locus the most likely causal variant across all traits and respective effect estimates (p15, line 428-430). We note that both variants are in strong LD ($R^2 > 0.85$) and MR effect estimates were very similar (odds ratio 4.85 vs 4.88). We have also rephrased the corresponding methods section and Figure 1 to clarify our approach.

Line 57: Sometimes "COVID-19 prognosis" is used, sometimes " COVID-19 severity" is used throughout the text.

R1 response 7 In response, we have harmonized the wording throughout the manuscript, also in accordance with comments raised by the second reviewer. We note that we have used the term 'prognosis' in its narrow clinical sense, describing any outcome of patients with COVID-19, irrespective of mild or severe.

Line 112: Supplementary Table 1 lacks the final colocalisation results showing that the 8 proteins mentioned are the top 8 results from colocalisation screenings. A supplemental table with all full 2375 proteins times 4 tested GWAS HGI sumstats results should be shown to demonstrate that only 8 proteins mentioned here had suggestive evidence ($p < 10^{-5}$).

R1 response 8 We provide the full list of colocalization results as novel Supplementary Table 2. We further realized during the review process, that our approach on prioritising candidate proteins was not well described and as a result have revised the entire section, including the application of various filtering steps to ensure robust and valid results (pages 14-15, lines 412-432).

Line 123: odds ratio: 4.85; 95%-CI: 2.65-8.89; p -value $< 3.1 \times 10^{-7}$ refers to variant rs7947771 (Suppl Table 2; also in the Abstract) but not to the discussed ELF5 variant rs766826 (rs766826 shown in Figure 3 and Discussion). I find it confusing when statistical effects are given sometimes for this variant and sometimes for that variant.

R1 response 8 We apologize for the inconsistency and like to refer to our answer above about the same topic (response 6).

Line 429: Number of proteins tested is sometimes 2,375 proteins, sometimes 2,345 as in Figure 2. Please correct.

R1 response 9 We apologise for the typo and have carefully revised all occurrences to reflect the actual number of proteins included, that is, those with a sufficiently conserved signal of the lead cis-pQTL in the overlapping set of SNPs.

Line 446: cis-pQTL colocalization p value threshold ($p < 10^{-5}$) needs to be further divided by 4 because 4 HGI GWAS sumstats were used for colocalization screening.

R1 response 10 *We understand that our description of the colocalization framework lacked details and have rephrased the corresponding section in the methods (pages 14-15, lines 412-432). The p-value threshold stated there is simply a filter for computational efficacy, since, in contrast to MR, statistical colocalisation needs at least suggestive evidence for genetic signals of both traits in the region. Our approach builds upon a successfully developed ‘colocalization first’ approach to identify proteins linked to diseases (Pietzner et al. 2021 Science), which addresses some of the short-comings of purely MR-based approaches in proteogenomic studies (Pietzner et al. 2021 Nature Communications).*

Line 131: Please indicate in Figure 2 which of the two original GWAS protein level sum statistics was used to find the result and whether it was significant in both studies. Please indicate in Figure 2 which P value is shown (from MR?) and which probability is meant here (posterior probability H3 or H4 from colocalization analysis?).

R1 response 11 *We implemented the proposed modifications in a revised version of Figure 2. None of the findings presented is seen for both platforms, and we have recently described in great detail the synergistic value of both proteomic technologies and that only triangulation with clinical or phenotypic information can identify biologically informative pQTLs (Pietzner et al. 2021 Nature Communications).*

Line 162: What does SNP rs35725681 in Figure 3 mean? The first 3 GWAS sum statistics shown in Figure 3 are exactly taken from the published HGI GWAS sum statistics, which should be mentioned in the legend.

R1 response 12 *We apologize for the mislabelling in Figure 3 and have corrected it in the revised version of the manuscript. We have further amended the figure legend to clearly state the origin of the GWAS summary statistics displayed.*

Line 142: „The lead cis-pQTL for ELF5, rs766826 (MAF=35.9%), is in strong linkage disequilibrium (LD; $r^2=0.81$) with a recently identified variant rs61882275 associated with severe COVID-19 in an independent study using whole genome sequencing³². The causal gene remained unidentified, ...“

Reading this, I have the impression that rs766826 and ELF5 are novel findings reported here for the first time and refine a genetic signal from a bioRxiv study of doi:10.1101/2021.09.02.21262965. However, rs766826 and ELF5 have been previously reported in the HGI GWAS study from which the GWAS summary statistics are taken here (see above). Please cite HGI bioRxiv.

R1 response 13 *We revised the corresponding section. We never intended to report rs766826 and ELF5 as ‘novel’ at this locus and now clearly refer to the COVID-19 HGI preprint (page 5, lines 130-131).*

Line 184: „We observed that ELF5 was almost exclusively expressed by different epithelial cells of the respiratory system“ is true only to these selected single cell datasets presented here (from the COVID-19 Atlas?). ELF5 is also expressed in breast tissue, bladder, skin and various other tissues, see <https://gtexportal.org/home/gene/ELF5>.

R1 response 14 *The reviewer is completely right, ELF5 is expressed in multiple tissues outside of the respiratory system. This statement was based on the finding that the underlying genetic signal, tagged by rs766826, is a cis-eQTL for ELF5 expression that is specific to the lung and not seen in other ELF5-expressing tissues, even at lenient thresholds (see Supplementary Figure 2). We have now rephrased the introduction of this section to clarify this better (page 7, lines 175-178).*

Further, we obtained single cell data from in-house studies and public resources (Durante et al. 2020), but not the COVID-19 atlas, and added this information to the revised version of the manuscript (page 7, line 177).

Line 201: Figure 4 shows previously published data, not data from new experiments. Please indicate in the legend.

R1 response 15 *We amended the legend of Figure 4 reporting the sources of each data set.*

Line 311: „We identified six not previously reported and replicated two known causal genes and“. Not correct, please cite HGI COVID-19 release 6.

R1 response 16 *We agree that our wording was misleading and an overstatement and now clearly acknowledge the findings from the COVID-19 HGI preprint (page 10, lines 277-278).*

Line 316: „We refine the association of genetic variation at ELF5 with severe COVID-19 to a single causal variant (rs766826) with high confidence (PP=99%)“. These PP results also apply to SNP rs766826 already as eQTL from the HIG release6 bioRxiv Paper, Supplementary Table 5, line 56 and 57 (<https://www.medrxiv.org/content/medrxiv/early/2021/11/11/2021.11.08.21265944/DC2/embed/media-2.xlsx?download=true>) and are therefore not new findings.

R1 response 17 *We thank the reviewer for pointing out the consistency between our hypercoloc-based fine-mapping approach and the fine-mapping done for cis-eQTLs from the GTEx consortium. We deleted the corresponding statement from the discussion.*

Line 335: " our scnRNAseq data provide strong evidence that ELF5 is expressed ...". Please reword as these single cell RNAseq data are not new and have already been published by others.

R1 response 18 *We rephrased the corresponding section to clearly state that we have used in-house as well as publicly available single cell/nuclei RNAseq data (page 7, lines 177-178).*

Line 461: "49 tissues of the 461 GTX v8 resource". The authors do not show that actual completely different tissues show expression for ELF5, why here the conclusion that ELF5 only for respiratory system, such as secretory and alveolar type 2 cells, when single cell data were studied only from these cells?

<https://gtexportal.org/home/gene/ELF5>

R1 response 19 *As outlined in our response to comment **R1 response 14**, we focused on scnRNAseq data from respiratory tissue for two reasons: 1) the tissue-specific evidence from eQTL colocalisation, and 2) as it is the most important tissue for severe COVID-19 patients suffering from acute respiratory distress syndrome.*

Line 742: Links to the published datasets cannot be found clearly. The GitHub page does not exist. Please provide exact links.

R1 response 20 *We provide links to the published data sets in the revised version of the manuscript along with a link to the GitHub repository containing the code used for the analysis.*

Line 742: The links to the published datasets do not point to the records. Also the GitHub page does not exist. Please provide correct links.

R1 response 21 *See our response to comment **R1 response 20**.*

Reviewer #2 (Remarks to the Author):

The authors report eight candidate proteins that are associated with COVID-19 outcomes. The authors leverage data on genetic variants and their association with protein levels in plasma, as this can be used to predict exposure of individuals to lifelong differences in protein levels in the tissue. The authors highlight a candidate protein ELF5 that has evidence for positive causal association with critical illness due to COVID-19, supported by further single cell RNA sequencing analysis and immunohistochemistry assays in lung showing specific expression in relevant cell types, and colocalization/co-expression with host viral entry proteins in cells susceptible to infection.

These approaches are important because they aim at disentangling the causal mediators of disease from the plethora of association results produced by large scale genome-wide association studies (GWAS) of host genetics of COVID-19. Better understanding of causal mechanisms behind infection and progression to severe disease can inform repurposing existing drugs and, in the long run, the creation of new ones that could benefit patients and prevent critical illness due to the virus, and finding biomarkers of disease progression. The analytical framework of the study is appropriate and the authors have undertaken a significant effort towards disentangling some of the host factors for COVID-19. I have questions regarding the methods, and some suggestions that I feel could improve the discussion of findings.

R2 response 1 *We thank the reviewer for kindly pointing out the strengths of our study.*

1) I think it would be best to list out all candidate proteins out in the beginning of the results section within the main text. Including them here makes it easier for the reader to

immediately know and refer back to these proteins when reading through the results/discussion instead of searching for them in the figures.

R2 response 2 *We followed the recommendation of the reviewer and now list all identified candidate mediators at the beginning of this section (page 5, lines 106-109).*

2) The authors use data for four COVID-19 outcomes released by the COVID-19 Host Genetics Initiative (HGI) that encompasses the largest sample sizes for any COVID-19 host genetic GWAS at this time. These four summary statistics were produced from analyses with highly overlapping samples (A2 included in B2 and B2 included in C2). Whilst running the analyses of this manuscript on all available summary statistics can be valuable in detail, the comparison of B1 analysis to the other outcomes is tricky due to the different definition of controls. Deeper discussion of these differences in ascertainment and what their implications could mean for the presented results would be needed. My suggestion would also be to keep B1 results in the supplementary tables but for ease of comparison across the other three case-definitions it would be clearer to show only analyses with the same definition of controls in the main text and Figure 2.

R2 response 3 *We agree with the reviewer that including so many slightly different and partially overlapping outcomes adds complexity to the presentation of results and we added a section to the discussion to clearly reflect possible insights that can be drawn from comparing across (page 12, lines 342-352). We further followed the suggestion of the reviewer and dropped B1 as an outcome in Figure 2 in the revised version of the manuscript.*

3) Figure 1 has mislabelled the A2 (shown as A1) and C2 (shown as C1), but I actually suggest removing them from the main text and figures as they are only meaningful to readers who know about the HGI analysis namings. These are fine in the methods and supplement. The naming of the outcomes should also be carefully checked throughout the manuscript for consistency as it is not always clear which outcome the authors refer to, e.g. HGI 'critical illness' (A2) definition seems to appear as 'severe COVID-19' and 'more severe COVID-19 prognosis'. I would suggest keeping to the original definitions when talking about the outcomes.

R2 response 3 *We thank the reviewer for pointing out this typo, which has now been corrected, and further followed the recommendation to align the wording when referring to any of the four outcomes, as also highlighted in our response to the **first reviewer, response 7**.*

4) It would be valuable to discuss in the manuscript how the COVID-19 outcome definitions used may differ from the actual COVID-19 phenotype that the genetic locus is affecting; deciphering whether a genetic locus found in the GWAS for a particular COVID-19 outcome is associated with susceptibility to viral entry and host infection or to the progression to severe disease after infection is difficult due to the nature of infectious disease and the phenotypes that can be collected. In addition, due to the overlap in samples between the HGI analyses (and ascertainments of samples, different overall sample sizes, etc), comparing the four COVID-19 GWAS summary statistics should be accompanied with discussion on how these can affect the interpretation. For example, a locus contributing to differences in

susceptibility to infection may be also present in critically ill vs. population outcome analysis, and a locus contributing to progression to more severe disease upon infection may be present in all cases vs. population outcome because of the inclusion of severe cases.

There have been some efforts to address this, e.g. the HGI has conducted in their update manuscript a Bayesian analysis to decipher which of the genetic loci is affecting infection susceptibility and which are associated with progression to severe disease. For example, in that analysis ELF5 locus is associated with disease severity and the ABO locus with infection susceptibility. This reviewer thinks that incorporating such information from published literature into the narrative of the manuscript could aid the interpretation of key figures (e.g. Figure 2) and conclusions of the article, e.g. do the MR findings support what has been previously reported? I think it is worthwhile to at least distinguish the measured outcomes from the underlying implication. E.g. in the introduction lines 97-98 when discussing outcomes the range is 'all positive cases' to confirmed COVID-19 positive and hospitalized with respiratory support or death', whilst the locus-specific effects might be 'increasing susceptibility to infection' or the 'risk of progression to severe disease once infected'.

R2 response 4 *We thank the reviewer for pointing out this important issue and have in response now rephrased the introduction (page 4, lines 96-97) and respective sections in the results (e.g., page 5, lines 110-115), and the discussion (page 12, line 342-352) to reflect the distinction of loci/proteins that might rather contribute to higher susceptibility to infection from those possibly underlying the progression to critical illness.*

5) Partially overlapping with the previous point, how much do the authors think the differences in GWAS sample sizes, overlapping samples between outcomes, coverage at each locus (how many studies out of all participating studies contributed data at that single instrument) can affect colocalization and MR in addition to the ancestry differences that the authors mention?

R2 response 5 *All these factors will have some effects on the results, and we have chosen a conservative analytical approach to safeguard against false-positive findings. Differing sample sizes will of course affect p-values, but testing for colocalization rather than following a traditional MR approach would provide evidence for a shared genetic signal even if evidence for association is below the genome-wide significance threshold, since the shape of the signal is tested, also mitigating confounding by LD. Colocalisation can further deal with sample overlap, while MR estimates are imprecise and tend towards the null if the outcome variable is measured with error, that is, not well defined.*

However, SNP coverage is an important issue and we addressed this using two filters: 1) we ensured that the cis-pQTL is sufficiently preserved in the overlap by testing whether the strongest pQTL in the overlapping set of SNPs is in strong LD ($R^2 > 0.8$) with the original regional sentinel (that is why some for some protein – outcome definitions reported SNPs slightly differ), and 2) included only SNPs from the COVID-19 HGI summary statistics with no less than 80% of the maximum sample size. We hope that the reviewer agrees that this is a rather conservative but robust strategy to prioritize findings. We rephrased the corresponding method section for clarification (pages 14-15, lines 411-431).

6) The authors discuss the ancestry differences between HGI summary statistics and the cis-pQTL data. Have the authors considered using the EUR-only summary statistics from HGI (I believe these are available for some of the population controls analyses)?

R2 response 6 *We and others have previously considered European-only statistics (Pietzner et al. 2021 Science, Klaric et al. 2021 MedRxiv), which produced inconsistent results and deliberately decided to make use of the trans-ethnic, and hence most powerful, analysis here. Although LD patterns are more likely to align between our UK-based protein statistics and the European sample, which is an advantage for colocalisation, the loss in statistical power outweighs the potential gain.*

We have now incorporated ancestry-specific fine-mapping at the *ELF5* locus, providing further support for the trans-ethnic transferability of the lead variant rs766826 (PP-Europeans = 66%, PP-African Ancestry= 83%).

7) How were the GWAS regions for coloc analysis defined? It was not mentioned in the methods, whether e.g. any P-value threshold was used to define regions around GWAS peaks. For example the G-CSF pQTL is colocalizing with a GWAS region that does not reach genome-wide significance. Could the authors comment on whether they have any concerns over the interpretation of coloc/MR results from GWAS loci that do not approach significance thresholds?

R2 response 7 *We apologize for the missing information and added a more detailed description to the corresponding methods section (pages 14-15, lines 411-431). Briefly, regions were chosen based on the gene body of protein-coding genes ($\pm 500\text{kb}$ each side) for all proteins covered on the SomaScan v4 assay or any of the twelve Olink panels. To be computationally efficient, we computed colocalization between protein abundances in blood and each of the four COVID-19 statistics only for regions for which there was at least suggestive evidence for a genetic signal for either trait ($p < 10^{-5}$).*

We agree with the reviewer that regions without strong evidence for statistical significance, e.g., $p < 5 \times 10^{-8}$, should be considered with caution, since we cannot rule out chance findings. However, by bringing together multiple orthogonal layers of data for each of the genes we highlight in detail, including CSF3, we ensure increased confidence of identifying a truly biologically relevant signal, even if this may not (yet) meet a statistical significance threshold defined based on a multiple testing burden. In other words, we increase confidence in true biological signals out of the wide range of subthreshold signals by integrating molecular information. Consider for instance the RAB2A finding, which has only been very recently identified as a novel hit for severe COVID-19 in a larger study (Pairo-Castineira et al. 2022 MedRxiv). We added a dedicated section to revised discussion highlighting possible concerns and arguments (page 12, line 353-363).

8) Did you consider any multiple testing correction for MR analysis? What did you consider as the threshold for significance?

R2 response 8 *We did not consider any multiple testing correction for MR analysis, since our analytical strategy is based around the establishment of a shared genetic signal, with MR*

*analysis secondary to this, mostly to provide directions of effect and to possibly quantify the effect. This strategy is based on experiences from our recent work on the proteogenomic convergence of human diseases (Pietzner et al. Science 2021) and cross-platform analysis (Pietzner et al. Nat Comms 2021) showing that proteins likely act via multiple routes beyond differences in blood levels. The protein altering variant prioritised at STFPD is another example, why colocalization, in our view, more reliably links proteins to diseases compared to MR only. We note that we observed for each protein candidate at least one significant effect after correcting for the number of tests done ($0.05/4*8$ tests; <0.0016), with ELF5 showing significant effects for all considered outcomes.*

9) Figure 2 shows the results of the MR analysis of association of predicted protein levels with risk of COVID-19 outcomes.

a) Current ordering is based on Gene/protein name alphabetical order. I would suggest ordering based on biology e.g. the above mentioned infection susceptibility and severity locus information or the statistical association strength of the COVID-19 loci, or something else. I suggest this because it could help see patterns in both panels, especially the right hand side one displaying shared/distinct signal.

b) The term 'shared genetic architecture' is often used when discussing results of genetic correlation analyses, and therefore it would be important to add a note to the legend and/or methods and main text that you are referring to colocalization probability PP.H4 as shared signal and PP.H3 as distinct signal.

c) Please report in the legend that the P-value is referring to MR analysis and the type of test so that this is easier to find in the methods.

d) The analysis results are plotted for each protein in reverse order (most severe COVID-19 outcome last) to how the results are displayed in Supplementary Table 2 (most severe outcome first). It would be easier to cross reference these if they were reported in the same order.

R2 response 9 *We thank the reviewer for those thoughtful comments and revised Figure 2 accordingly. We further rephrased the term 'shared genetic architecture' to 'shared genetic signal' for clarification.*

10) Please define on line 158 what you mean by 'unrelated', i.e. was a P-value for a particular test not significant?

R2 response 10 *We rephrased this section to clearly indicate that 'unrelated' referred to the absence of evidence for colocalization between plasma levels of catalase and any COVID-19 outcome based on multi-trait colocalization (page 6, line 150).*

11) Authors refer to putative causal variants in the manuscript but it is unclear which method was used to define causal variants (i.e. was separate fine-mapping performed or was this run as part of coloc?). If this was given directly as the colocalization output, it would be useful to mention this in the methods and report the posterior probabilities for each SNP in Supplementary Table 2 (SNP.PP.H4). Also at times different variants are referred to as candidate causal variant at a locus, e.g. at the G-CSF locus the Supplementary Figure 1 refers to rs4795412 whereas Supplementary Table 2 contains variants rs3826331 and rs8081692. What is behind the difference?

R2 response 11 *We apologize for the missing information and added it to the revised version of the manuscript (page 15, line 432-443). Briefly, we identified putative causal variants based on the output from multi-trait colocalization ('hyprcoloc') and not pairwise colocalization ('coloc'). The inconsistency among reported variants for G-CSF and other protein examples, was due to the varying overlap of SNPs among the four different COVID-19 outcome definitions and we reported simply the strongest pQTL for each overlapping set. This also meant that the overlap for multi-trait colocalization was smallest. However, all SNPs are in strong LD ($r^2 > 0.8$) and hence likely tag the same underlying genetic signal. We extended the method section to describe these inconsistencies more clearly (page 15, lines 427-431).*

12) The authors' findings and previous literature points to AT2 cells as being important for severe COVID-19 disease outcomes. It would be helpful to orient the reader to lung biology by adding a phrase or two about the role of these cells in healthy tissue (or also in disease) when these cells are first described in the main text.

R2 response 12 *We followed this very helpful suggestion and added a small section on the role of AT2 cells to the revised version of the manuscript (page 7, line 190-194).*

13) Figure 4:

a) Panel C: Can ACE expression be shown as a gradient similarly as ELF5 in panel B?

R2 response 13a *We appreciate that having a similar colour gradient for ACE2 as for ELF5 would be helpful. However, similar to many other previous studies, ACE2 expression is usually so low, that no colour gradient would give distinguishable results. The field has rather taken the approach to mark ACE2 transcript⁺ cells as a surrogate, which is far more informative to identify cells expressing this gene (see for example, Sungnak et al. 2020 Nature Medicine, Muss et al. Nature Medicine 2021 or Ziegler et al. 2020 Cell).*

b) Panel D: Could you also add to the legend or methods a note how the genes on the right hand side panel (TMPRSS2, FURIN, NRP1, NRP2) were chosen for display as this is not obvious with the exception of the TMPRSS2?

R2 response 13b *We thank the reviewer for pointing out this ambiguity on the rationale behind the display of host-factor genes in Figure 4D. Following this helpful suggestion, we have added an explanation to the figure legend*

14) The authors highlight an intriguing possible explanation for the tissue specificity of ELF5 expression. It would be great to outline the crucial region more clearly in Supplementary Figure 4, as it is a bit difficult to interpret without more direction. Should the reader consider the lung open chromatin regions as a whole (all highlighted area) or the block just to the right of the SNP of interest? The tissues could also be ordered on the y-axis by expression level in the tissue from highest to lowest to make the comparison between lung and other highly expressing tissues mentioned in the main text easier.

R2 response 14 *We thank the reviewer for this helpful advice and adapted Supplementary Figure 4 as suggested.*

15) Regarding Figure 6 and for phenome-wide colocalization analysis at CSF3 locus:

a) I'm wondering whether it would be more informative to show the non-standardized z-scores, so as to get an impression of the relative effects of variants in this locus on the different phenotypes? Currently from what I understand the figure mainly shows that variants in higher LD with the candidate causal variant also have stronger association with the trait as is expected based on the definition of LD. For that reason also most of the molecular traits in the figure have a similar LD/zscore pattern as they presumably come from the same study.

b) Additionally for the COVID-19 traits, the z-scores seem to distribute differently in the group of variants in high LD to the candidate variant. Can this be explained by mixed ancestry included in the meta-analysis?

c) Methods should include how LD was estimated here.

d) Was any multiple testing correction applied to phenome wide colocalization analysis?

e) It would make sense to include in brackets in the main text that the G-CSF is in CSF3 locus to better link this figure to the main text (as currently one has to look at Figure 1 to figure this out).

f) The legend title says "GCSF3" - is this a typo? Also "gene variant" is probably a typo and to be replaced with "genetic variant" as it is not in a gene.

g) The ordering of the traits on the figure should be grouped more logically e.g. all neutrophil phenotypes together (or even better, select a few representative ones).

R2 response 15 *We followed the helpful recommendations of the reviewer and now provide stacked locuszoom plots for selected and biologically grouped traits at the CSF3 locus. We further added the requested information to the text/figure/legend and apologize for the typo. The less obviously 'shaped' signal for COVID-19 outcomes at this locus might be a combination of both, low statistical power, and hence more noisy individual SNP estimates, and non-matching LD patterns. We note that the same locus has been identified in trans-ethnic analysis of white blood cell counts (Chen et al. 2020 Cell).*

We did not apply any formal multiple testing correction to our phenome-wide colocalization approach and rather sought for orthogonal validation at loci/for proteins with only suggestive statistical evidence, such as for G-CSF by showing convergence with blood cell phenotypes mimicking the biological function of the protein and the clinical presentation of critical illness in COVID-19. We would generally argue that cumulative biological evidence should always outweigh statistical arguments but included an additional limitation sentence in the revised version of the manuscript to acknowledge this fact explicitly (page 12, line 353-363).

16) Supplementary Table 4 does not include the COVID-19 outcomes plotted in Figure 6, so these should be added.

R2 response 16 *We added COVID-19 outcomes to ST4 now ST5.*

17) Something that I found confusing at first was that for all stacked colocalization plots, it seems that sometimes the lighter filled points (lower r^2) can look very dark because they are clustered together and the dark borders make them look like a dark point. It made the figure look like some high r^2 variants have low P-values. Would it be possible to modify the border colour or alternatively narrow in on the locus to spread the points out more to make sure the light colour comes out?

R2 response 17 *We followed the very helpful suggestion of the reviewer and revised the corresponding plots.*

18) Supplementary Figure 3: Should be noted in the legend that the bar represents where this other variant rs35725681 is. It could be also worth highlighting the likely causal variant for ELF5 pQTL to give more context to this region. Maybe highlight both variants with diamonds to show which variant it is instead of the bar?

R2 response 18 *We amended the figure legend of SF3 and have now labelled both variants in the plot.*

19) Lines 175-182: Curiously, a similar counterintuitive effect has been seen for IPF associated MUC5B locus by Fadista et al (<https://doi.org/10.1016/j.ebiom.2021.103277>).

R2 response 19 *We thank the reviewer for pointing out this interesting study and incorporated it in the discussion of our findings (page 7, line 171-173).*

20) Typo on line 486? Seems like it should be referring to Supplementary Figure 12.

R2 response 20 *Thank you for spotting this typo, which has been corrected in the revised version of the manuscript.*

21) Discussion:

a) Repetition of content on lines 311-315 and 324-327 regarding results, and on lines 322 and 331 about Ets transcription factor family.

R2 response 21a *We revised the corresponding section to minimize redundancy (page 10, lines 273-278).*

b) On lines 393-397: Could part of the interpretation of MR effect estimate being low also be that the GWAS effect points to infection susceptibility (instead of disease progression) is on its own relatively modest?

R2 response 21b *Very modest effects might very well contribute to the low effect estimates and we added this notion to this section (page 13, lines 381-382).*

c) Line 334: Statement about embryonic lethal mouse is missing a reference.

R2 response 22c *We have now added a reference to the embryonic lethal mouse model.*

d) Lines 342-345: Is this missing a reference to mouse study? Can you specify how the findings partly align as this phrase is a bit difficult to interpret.

R2 response 22d *We added the relevant reference to the mouse model and rephrased the sentence for clarity, that is, that ELF5 expression positively correlates with viral entry proteins in mouse and humans (page 10, lines 297-300).*

e) Line 350: Unclear what “if any” is referring to.

R2 response 22e *We have clarified our statement regarding this (pages 10-11, lines 304-305).*

f) Line 352: Likely authors mean “infection symptoms” as “infectious symptoms” implies anosmia is infectious. Also, what is the symptom highly specific for?

R2 response 22f *We thank the reviewer for pointing out this error; we have deleted this part of the sentence as it didn't add much to the discussion.*

g) Lines 366-369: The wording here is a bit speculative as the authors say ELF5 is likely one of many contributors but also only possibly contributing to anosmia, so I recommend rephrasing.

R2 response 22g *We followed the recommendation of the reviewer and rephrased the corresponding section to clearly differentiate both findings, since the evidence for anosmia is currently weaker compared to a possible role of ELF5 in severe COVID-19 (pages 11, line 321-322).*

h) Lines 409-410 again referring back to my comments about susceptibility to infection/progression to severe disease, I would add specifically the proteins here that have clearer evidence of modulating disease severity and not infection susceptibility if this is the desired message, or mention that some loci have more evidence of being associated with infection susceptibility.

R2 response 22h *We followed the helpful suggestion from the reviewer and rephrased the section for clarity (page 12, lines 353-363).*

i) ELF5 has been highlighted as a candidate gene in the region before by two GWAS mentioned in the text, and HGI paper which mentions ELF5 fills the criteria "closest gene to the lead SNP, and an eGene i.e. fine-mapped cis-eQTL variant PIP > 0.1 in GTEx Lung, in LD with a COVID-19 lead variant $r^2 > 0.6$ ". Whilst this study is the first to follow up with pQTL and RNA analyses and staining, this reviewer is leaning towards thinking the protein candidate is perhaps not 'novel' but the current study significantly strengthens the evidence for this gene product.

R2 response 22i *We agree with the reviewer that previous studies already provided evidence that ELF5 might be the candidate causal gene at this locus. As also outlined in our **response 1***

to reviewer #1, we thoroughly revised the entire manuscript, to make it clear what our contribution is and what has already been reported.

Methods

22) Please describe in the methods you are using COVID-19 summary statistics versions with all ancestries and 23andMe excluded. Please also include in the methods and supplementary table legends that you are using hg19 genome build versions and the same for cis-pQTL.

R2 response 22 We apologize for the missing information and added those to the revised version of the manuscript (page 14, lines 405-406).

23) A description of the method and analysis for “multi-trait colocalization” analysis mentioned on lines 126 and 293 and referred to as Reference number 31 seems to be missing from the methods section.

R2 response 23 We added a paragraph on multi-trait colocalization to the revised version of the manuscript (page 15, lines 432-443).

24) I would recommend breaking down the “Single-cell/nucleus RNA sequencing” section into two separate parts covering the quality control and then analyses. Also good to refer in the methods to the same analysis name as in the main text, e.g. main text talks about cell-type specific gene set enrichment analysis, whilst the methods talks about gene set enrichment analysis. This would make it easier for the reader to confidently find the right section in the methods.

R2 response 24 We followed the reviewer’s suggestion to split the methods section regarding the single-cell/nucleus sequencing for better readability (pages 16-17, lines 480-500). We have harmonized the use of cell-specific pathway enrichment and gene set enrichment analysis within the manuscript.

References:

25) References #14 and #30 to be fixed, the author is the consortium 'COVID-19 Host Genetics Initiative'

R2 response 25 Thank you for the advice, we have now fixed the reference.

26) Lines 78-79 of the main text: I would add the Zhou and Downes paper references where the genes are mentioned in this sentence to help the reader identify which study points to which gene. Authors could also consider Kasela et al PMID: 34425859 when discussing chr3 locus and SLC6A20

R2 response 26 We incorporated this helpful suggestion in the revised version of the manuscript (page 3, line 79).

REVIEWER COMMENTS

Reviewer #1 (Remarks to the Author):

I thank the authors for revising the points I raised. Many comments and questions have been answered. However, I have a different view on the colocalization analysis results and I think the authors should improve some methodologically very important points of the presented colocalization filtering analysis and candidate selection process and should improve Suppl Table 2. The colocalization results as well as their interpretation are not reproducible (and in my opinion also partially incorrect or incomplete) based on the available data and method descriptions. Only 2 (BGAT and OAS1) of the 8 proteins reported as causal candidates meet accepted statistical criteria for genome-wide colocalization analyses. I ask for a response to the points below so that the reader understands the (correctness of the) approach.

Major points

R1 response 8 and R1 response 10

I thank for the post submission of the results of the colocalization analysis in the form of Suppl Table 2, in addition to the more detailed method description, of the four genome-wide colocalization analyses between the genome-wide SNP-protein summary statistics and the four genome-wide HGI GWAS SNP-disease sum statistics. I would have expected the main results of this analysis as one main table in the manuscript (but now with only the genome-wide significant results of BGAT and OAS1 with protein GWAS p-values and COVID19 GWAS p-values and PP_H4 values), as these analyses and results are the heart of the study and all functional protein follow-up analyses are based on their significance.

The authors write "We understand that our description of the colocalization framework lacked details and have rephrased the corresponding section in the methods (pages 14-15, lines 412-432). The p-value threshold stated there is simply a filter for computational efficacy, since, in contrast to MR, statistical colocalisation needs at least suggestive evidence for genetic signals of both traits in the region. Our approach builds upon a successfully developed 'colocalization first' approach to identify proteins linked to diseases (Pietzner et al. 2021 Science), which addresses some of the short-comings of purely MR-based approaches in proteogenomic studies (Pietzner et al. 2021 Nature Communications)."

Unfortunately, I don't understand the merits or improvement of this "colocalization first" optimization approach. It allows "only testing regions with at least suggestive evidence ($p < 10^{-5}$) for either the plasma protein or COVID-19-related outcomes" (lines 429-430). This obviously leads to an inflation of the coloc test results and a large number of (possibly false positive) protein GWAS and disease GWAS SNPs colocalization results (although $PP_{H4} > 0.8$), since results are only nominal significant but not significant after correction for multiple testing in the individual protein GWAS (which must include multiple testing correction for 1 million SNP variants and also number of tested proteins) and the HGI COVID19 binary GWAS summary statistics.

There is a clear difference here between the coloc analysis performed here and the coloc analysis presented in Pietzner 2021 Science. In Pietzner 2021 Science, only 10,674 study genome-wide significant variant protein-target associations with a GWAS $P < 5 \times 10^{-8} / 3892 \text{ proteins} = 1.004 \times 10^{-11}$ were defined as pQTLs and tested for colocalization (including strong proxy SNPs with $r^2 > 0.8$) with established genome-wide significant GWAS catalog disease-associated SNPs ($P < 5 \times 10^{-8}$) from genome-wide GWAS catalog summary statistics. In this study, the protein GWAS data were already combined with the four different HGI COVID-19 summary stats by statistical colocalization and the Coloc program and the authors wrote "We replicated the previously reported candidate genes ABO and OAS1 (43) (fig. S8), both of which showed consistent evidence across these different outcome

definitions.”

This is a statistically robust approach because it is generally accepted that for colocalization analyses between genome-wide SNP-protein and SNP-disease summary statistics, genome-wide GWAS and genome-wide study-wide protein GWAS thresholds must apply to both summary statistics. For example, Gudjonsson and colleagues (<https://pubmed.ncbi.nlm.nih.gov/35078996/>) had the following accepted approach:

“For each trait, significant loci were defined by identifying all genome-wide variants ($P < 5 \times 10^{-8}$) at least 500 kb apart, defining a flanking region of 1Mb around each lead variant, and finally merging overlapping regions. For each GWAS locus, all SOMAmers with a study-wide significant association (cis or trans) within the given region were tested for colocalization, if at least 50 SNPs in the region had complete information from both trait and protein GWAS and the overlapping set of SNPs included at least one SNP with a genome-wide significant ($P < 5 \times 10^{-8}$) phenotype association and at least one SNP with a study-wide significant ($P < 1.046 \times 10^{-11}$) protein association. When the MAF was not available for a given GWAS, the 1000 Genomes EUR MAF was used instead. Colocalization analysis was performed with coloc (v.3.2-1), using the coloc.abf function with default priors. In a secondary analysis we repeated the analysis with a more stringent prior selection, $p_{12} = 5 \times 10^{-6}$, as recently proposed (Wallace 2020). High and medium colocalization support was defined as $PP.H4 > 0.8$ and $PP.H4 > 0.5$, respectively.” The results can be found in data table 10 of Gudjonsson and colleagues, see <https://www.nature.com/articles/s41467-021-27850-z#Sec18>

The reasons for study-specific genome-wide omics GWAS thresholds are that with such a seemingly stringent genome-wide threshold alone, in fact still in Gudjonsson et al for example, “269,637 variants exhibited study-wide significant associations ($P < 5 \times 10^{-8}$ / 4,782 SOMAmers = 1.046×10^{-11}), with 2112 unique proteins, dubbed protein quantitative trait loci (pQTLs).” Another example: Horowitz and colleagues (<https://pubmed.ncbi.nlm.nih.gov/33619501/>) considered in their COVID19 GWAS only genome-wide significant ($p < 5 \times 10^{-8}$) COVID19 GWAS variants (or variants with high LD $r^2 > 0.8$) and eQTLs associated with gene expression at a $P < 2.5 \times 10^{-9}$ from the original eQTL summary stats for their colocalization analyses.

The newly reported causal proteins HSP40, RAB2A, and NUDT5 mentioned in lines 108, 284 and Abstract, not pQTL genome-wide significant ($p < 1.004 \times 10^{-11}$) except for NUDT5 in the protein GWAS sum stats with their lead variants, these variants do not even show a suggestive significant GWAS P-value of $p < 10^{-5}$ in any of the 4 HGI COVID19 GWAS sumstats shown (see HGI GWAS p-values 0.00012872 0.00014677 and 0.00013432 in rows 9, 31, 47 in Suppl Table 2). (Possibly missing in Suppl Table 2 are the slightly better proxy variants from the HIG GWAS sum stats with $p < 10^{-5}$, see comment below).

Also in Suppl Table 3 (erroneously also shown in Figure 3, see comment below) it is also very clear that the HGI GWAS variants/P-values for HSP40, RAB2A, and NUDT5 show almost no association P-value signal, despite the fact that the HGI COVID19 data include huge sample sizes.

Therefore, I would advise against calling the HSP40, RAB2A, and NUDT5 novel causal COVID19 candidate genes/proteins or COVID19 risk loci in in Abstract, Results and Discussion, because there is insufficient statistical signal from the COVID19 summary statistics. Only genome-wide significant SNPs from the HGI COVID19 data justify the labeling of a risk locus.

Only the BGAT (ABO) and OAS1 proteins should be listed in a new main Table 1, as shown, for example, in Table 1 from Yao et al. <https://pubmed.ncbi.nlm.nih.gov/30111768/>, as the text in lines 108-112 is so short and does not describe this fact that only these two genes/proteins meet the accepted genome-wide colocalization significance criteria from both GWAS summary statistics, as used in other publications. ELF5 can also be mentioned here (possibly also in a main Table 1) as the most promising new candidate without meeting the accepted genome-wide colocalization significance criteria, as shown in Figure 3, with genome-wide significance in the HGI COVID19 data, but unfortunately no pQTL study-related genome-wide significance. All other 5 mentioned "causal" protein candidates (SFTPD, CSF3, HSP40, RAB2A, NUDT5) do not show genome-wide significance in any of

the 4 HGI COVID19 GWAS data (sometimes not even $p < 10^{-4}$) nor in the pQTL GWAS data and therefore do not fit the description of "causal candidate" at all, as the locus itself does not show significance for COVID19.

Another question: After the previous candidate selection, why doesn't the HINT1 gene (line 12 in Suppl Table 2) qualify as another suggestive gene/protein (instead of HSP40, RAB2A and NUDT5), with a much smaller $P = 2.14 \times 10^{-14}$ for protein GWAS, PP_H4=84.21% and some "more" suggestive HGI COVID GWAS $P = 7.80 \times 10^{-5}$? And what about the FAS gene with protein GWAS $P = 5.92 \times 10^{-131}$, PP_H4=93.07%, HGI COVID GWAS $P = 1.04 \times 10^{-4}$ in Suppl Table 2? For clarity, it might be helpful dividing Suppl Table 2 into the 4 coloc sub-result tables for the 4 HGI GWAS sumstats coloc analyses and then sort the results by significance, with significant genes like BGAT and OAS1 at the top.

For the same reason, all genes except BGAT, OAS1 and ELF5 should be removed from Figure 4, as there might be statistically "better" candidates such as HINT1 or Fas from Suppl Table 2, which were not mentioned here. CSF3 in Figure 6 should not (yet) be mentioned as a COVID19 locus risk because not genome-wide significant, with the indication that this locus may yet be detected in the future in an even larger study than that of the HGI to date.

For better reproducibility, please write in the methods section for lines 420-441 how many genetic variants remain after each filter step.

Line 422: For the coloc program analysis, please report which prior probabilities p_1, p_2, p_{12} were used because number of variants with $PP_H4 > 0.8$ depends also on the value of prior p_{12} . Please indicate from where the MAFs originate, which were used as input per SNP. It has been shown that the default value of $1e^{-5}$ for the p_{12} prior of the coloc program shifts protein-phenotype pairs often too easily above the threshold of 0.8. (Supplementary Fig. 12 (A); Gudjonsson et al; <https://pubmed.ncbi.nlm.nih.gov/35078996/>). Please repeat also the analysis with a more stringent prior selection, $p_{12} = 5 \times 10^{-6}$, as recently proposed (Wallace 2020 <https://pubmed.ncbi.nlm.nih.gov/32310995/>) and as performed in Gudjonsson et al., since $p_{12} = 10e^{-5}$ is overly liberal (Wallace 2020). Please use prior probabilities of COVID19 case-control association e.g. $p_1 = 23/2000000 = 1.15e^{-5}$ (i.e. number of gws case-control hits / number of common SNPs (about 2, 000, 000), see rules from Wallace 2020; i.e. currently 23 gws loci from COVID19 HGI sumstats). Prior p_2 should perhaps also be set at $1.15e^{-5}$ rather than $1e^{-4}$, because for SNP-protein GWAS, the study-based genome-wide significance threshold is far below that of case-control GWAS because there are thousands of protein tests genome-wide, e.g., $1.004 \times 10e^{-11}$ in Pietzner 2021 Science and 1.046×10^{-11} in Gudjonsson et al.

Please make all pQTL summary statistics mentioned in lines 401-411 and used for colocalization analysis publicly available for download with a link in Section "DATA AVAILABILITY" (line 536-541).

Can the authors please provide quantile-quantile plots and inflation factor calculations for the pQTL GWAS summary statistics used here? (because suggestive signals were examined) (Similar to the publication by Gudjonsson and colleagues <https://pubmed.ncbi.nlm.nih.gov/35078996/>)

Line 435: The number "1,245 unique protein targets with trustworthy colocalization results" cannot be identified or counted (when counting proteins) in Suppl Table 2.

Regarding lines 436-441: Please add the different SNP variant IDs for the 4 HGI COVID-19 sum statistics to show the best proxy SNP for each of the COVID19 sum statistics colocalized with the best SNP from the protein GWAS. Please also include LD values between all SNP-SNP pairs from individual protein and COVID-19 sum comparisons. Otherwise, results are unfortunately not reproducible or usable for other groups. The authors can be guided in the preparation of Suppl Table 2 by the Data Table 10 by Gudjonsson and colleagues, see <https://www.nature.com/articles/s41467-021-27850-z#Sec18>

The sentence in line 432 is not clear to me, please specify: "we further tracked whether the lead cis-pQTL was sufficiently preserved in the set of SNPs common to the protein and COVID-19 summary statistics". Please explain what this means.

Unfortunately, the answer "R1 response 11" is not clear to me. Why are none of the signals from Figure 2 at least nominally significant and replicable across the two pQTLs GWAS summary statistics? Is there at least the same trend in terms of effect sizes? How to be sure that they are not false-positive pQTL associations then? Do the pQTL signals come only from one pQTL sumstats file but not from the other? Please explain.

Regarding "R1 response 11": If the P-values in Figure 2 are from Mendelian randomization analysis, the p-value threshold for determining significance must be defined somewhere. To me it looks more like the P-values shown in Figure 2 (contrary to the Figure 2 description) are not MR P-values but the P-values from the HGI COVID19 SNP Sumstats, see also Supp Table 2, columns "Pvalue". The same P-values from Figure 2 are also listed in Suppl Table 3 as "pval.mr" (legend for column names is missing). I have looked in the HGI COVID19 data myself and these are all HGI COVID19 SNP P-values and therefore cannot be MR P values. So is there something wrong with the P-values of the MR analyses provided in Figure 2 and Suppl Table 3? Can the authors please check once the MR P values with another alternative method, e.g. Summary-data-based Mendelian Randomization tool SMR (Zhu 2016 Nat Genet. PubmedID <https://pubmed.ncbi.nlm.nih.gov/27019110/>) or MR-Egger (Bowden 2015 Int J Epidemiol <https://pubmed.ncbi.nlm.nih.gov/26050253/>) since I am unfortunately not familiar with the Wald ratio MR method of Lawlor 2008, which is mentioned only by two sentences in the method section?

Minor points

Regarding the answer "R1 answer 4", the text in lines 133-134 still lacks the clear indication that the pQTL SNP rs766826 has also already been identified as eQTL SNP rs766826 by COVID19 HGI publication #15, which of course should not be a requirement here, but further emphasizes the validation at the protein level here.

New Lines 112/113: "All genetic variants possibly linking proteins to COVID-19 had highest effect estimates for severe COVID-19". Which effect estimates of which analysis are meant here? From the case-control effect estimates of the HGI GWAS data (Suppl Table 3) or are we already referring to the Mendelian randomization analysis (also results in Suppl Table 3)? If the former, then please provide case-control effect estimates of SNPs in ORs in Suppl Table 3 (and also new main Table 1), otherwise it is difficult to understand the statement "highest effect estimates" when showing beta values instead of ORs for case-control analyses.

Reviewer #2 (Remarks to the Author):

I thank the authors for their thorough responses. All my questions have been addressed, and important discussion and methodological detail have been added to the manuscript. I have no further questions.

REVIEWER COMMENTS

Reviewer #1 (Remarks to the Author):

I thank the authors for revising the points I raised. Many comments and questions have been answered. However, I have a different view on the colocalization analysis results and I think the authors should improve some methodologically very important points of the presented colocalization filtering analysis and candidate selection process and should improve Suppl Table 2. The colocalization results as well as their interpretation are not reproducible (and in my opinion also partially incorrect or incomplete) based on the available data and method descriptions. Only 2 (BGAT and OAS1) of the 8 proteins reported as causal candidates meet accepted statistical criteria for genome-wide colocalization analyses. I ask for a response to the points below so that the reader understands the (correctness of the) approach.

We thank the reviewer for the critical revision of our paper and the opportunity to clarify our statistical approach, which we agree will help to ensure reproducibility. Specific responses to the points raised can be found below.

Major points

R1 response 8 and R1 response 10

I thank for the post submission of the results of the colocalization analysis in the form of Suppl Table 2, in addition to the more detailed method description, of the four genome-wide colocalization analyses between the genome-wide SNP-protein summary statistics and the four genome-wide HGI GWAS SNP-disease sum statistics. I would have expected the main results of this analysis as one main table in the manuscript (but now with only the genome-wide significant results of BGAT and OAS1 with protein GWAS p-values and COVID19 GWAS p-values and PP_H4 values), as these analyses and results are the heart of the study and all functional protein follow-up analyses are based on their significance.

***R1 response 1** We followed this recommendation and now provide a new table 1 in the revised version of manuscript. To clearly indicate the statistical stringency of our prioritized examples, we further introduced a tier system and rank examples accordingly. Briefly, we distinguish candidate proteins (i.e., PP H4 > 80% and LD regional sentinels $r^2 > 0.8$) on the following basis: 1 – genome-wide significant ($p < 5 \times 10^{-8}$) in protein and COVID-19 summary statistics; 2 – genome-wide significant in either protein or COVID-19 statistics; and 3 – suggestive candidates with subthreshold findings in both (p5, line102-110) and put most emphasis on the first two categories.*

The authors write “We understand that our description of the colocalization framework lacked details and have rephrased the corresponding section in the methods (pages 14-15, lines 412-432). The p-value threshold stated there is simply a filter for computationally efficacy, since, in contrast to MR, statistical colocalisation needs at least suggestive evidence for genetic signals of both traits in the region. Our approach builds upon a successfully developed ‘colocalization first’ approach to identify proteins linked to diseases (Pietzner et al.

2021 Science), which addresses some of the short-comings of purely MR-based approaches in proteogenomic studies (Pietzner et al. 2021 Nature Communications).”

Unfortunately, I don't understand the merits or improvement of this "colocalization first" optimization approach. It allows “only testing regions with at least suggestive evidence ($p < 10^{-5}$) for either the plasma protein or COVID-19-related outcomes” (lines 429-430). This obviously leads to an inflation of the coloc test results and a large number of (possibly false positive) protein GWAS and disease GWAS SNPs colocalization results (although $PP_H4 > 0.8$), since results are only nominal significant but not significant after correction for multiple testing in the individual protein GWAS (which must include multiple testing correction for 1 million SNP variants and also number of tested proteins) and the HGI COVID19 binary GWAS summary statistics.

R1 response 2 We agree that we should have explained this better, rather than referring to our previous work, especially since our approach did not exactly follow what has previously been published. We now explain our rationale in more detail (p14, lines 417-421): our strategy was to first establish a shared genetic signal between a protein and an outcome as the basis to prioritise examples for formal MR analyses. The importance of this has been highlighted by Zuber et al. 2022 AJHG specifically for the context of single variant MR, in other words, without this prioritisation step the false-discovery rate for single variant cis-MR is has been reported to be unacceptably high. As outlined above, we further introduced a tier system to clearly label candidate proteins based on statistical significance.

There is a clear difference here between the coloc analysis performed here and the coloc analysis presented in Pietzner 2021 Science. In Pietzner 2021 Science, only 10,674 study genome-wide significant variant protein-target associations with a GWAS $P < 5 \times 10^{-8} / 3892$ proteins = 1.004×10^{-11} were defined as pQTLs and tested for colocalization (including strong proxy SNPs with $r^2 > 0.8$) with established genome-wide significant GWAS catalog disease-associated SNPs ($P < 5 \times 10^{-8}$) from genome-wide GWAS catalog summary statistics. In this study, the protein GWAS data were already combined with the four different HGI COVID-19 summary stats by statistical colocalization and the Coloc program and the authors wrote “We replicated the previously reported candidate genes ABO and OAS1 (43) (fig. S8), both of which showed consistent evidence across these different outcome definitions.”

This is a statistically robust approach because it is generally accepted that for colocalization analyses between genome-wide SNP-protein and SNP-disease summary statistics, genome-wide GWAS and genome-wide study-wide protein GWAS thresholds must apply to both summary statistics. For example, Gudjonsson and colleagues (<https://pubmed.ncbi.nlm.nih.gov/35078996/>) had the following accepted approach:

“For each trait, significant loci were defined by identifying all genome-wide variants ($P < 5 \times 10^{-8}$) at least 500 kb apart, defining a flanking region of 1Mb around each lead variant, and finally merging overlapping regions. For each GWAS locus, all SNPs with a study-wide significant association (cis or trans) within the given region were tested for colocalization, if at least 50 SNPs in the region had complete information from both trait and protein GWAS and the overlapping set of SNPs included at least one SNP with a genome-wide significant ($P < 5 \times 10^{-8}$) phenotype association and at least one SNP with a study-wide significant ($P < 1.046 \times 10^{-11}$) protein association. When the MAF was not available for a given GWAS, the 1000 Genomes EUR MAF was used instead. Colocalization analysis was performed with coloc (v.3.2-

1), using the coloc.abf function with default priors. In a secondary analysis we repeated the analysis with a more stringent prior selection, $p_{12} = 5 \times 10^{-6}$, as recently proposed (Wallace 2020). High and medium colocalization support was defined as $PP.H4 > 0.8$ and $PP.H4 > 0.5$, respectively.” The results can be found in data table 10 of Gudjonsson and colleagues, see <https://www.nature.com/articles/s41467-021-27850-z#Sec18>

R1 response 3 We followed the helpful suggestion of the reviewer to investigate the influence of prior specification on our results and established a comprehensive sensitivity analysis by testing a grid of successively more stringent priors for all presented candidate proteins. Briefly, all candidate proteins showed strong ($PP.H4 > 85\%$) evidence for colocalization in the vast majority of prior settings ($>95\%$) with very few exceptions. We added these results to the revised version of the manuscript (p5, line 108-109).

The reasons for study-specific genome-wide omics GWAS thresholds are that with such a seemingly stringent genome-wide threshold alone, in fact still in Gudjonsson et al for example, “269,637 variants exhibited study-wide significant associations ($P < 5 \times 10^{-8}/4,782$ SOMAmers = 1.046×10^{-11}), with 2112 unique proteins, dubbed protein quantitative trait loci (pQTLs).” Another example: Horowitz and colleagues (<https://pubmed.ncbi.nlm.nih.gov/33619501/>) considered in their COVID19 GWAS only genome-wide significant ($p < 5 \times 10^{-8}$) COVID19 GWAS variants (or variants with high LD $r^2 > 0.8$) and eQTLs associated with gene expression at a $P < 2.5 \times 10^{-9}$ from the original eQTL summary stats for their colocalization analyses.

The newly reported causal proteins HSP40, RAB2A, and NUDT5 mentioned in lines 108, 284 and Abstract, not pQTL genome-wide significant ($p < 1.004 \times 10^{-11}$) except for NUDT5 in the protein GWAS sum stats with their lead variants, these variants do not even show a suggestive significant GWAS P-value of $p < 10^{-5}$ in any of the 4 HGI COVID19 GWAS sumstats shown (see HGI GWAS p-values 0.00012872 0.00014677 and 0.00013432 in rows 9, 31, 47 in Suppl Table 2). (Possibly missing in Suppl Table 2 are the slightly better proxy variants from the HIG GWAS sum stats with $p < 10^{-5}$, see comment below).

Also in Suppl Table 3 (erroneously also shown in Figure 3, see comment below) it is also very clear that the HGI GWAS variants/Pvalues for HSP40, RAB2A, and NUDT5 show almost no association P-value signal, despite the fact that the HGI COVID19 data include huge sample sizes.

Therefore, I would advise against calling the HSP40, RAB2A, and NUDT5 novel causal COVID19 candidate genes/proteins or COVID19 risk loci in in Abstract, Results and Discussion, because there is insufficient statistical signal from the COVID19 summary statistics. Only genome-wide significant SNPs from the HGI COVID19 data justify the labeling of a risk locus.

R1 response 4 In line with our responses to earlier comments, we have now adopted the suggested strategy and clearly categorized and labelled protein mediators into tiers, focussing on groups 1 and 2 as proposed. We have rephrased the corresponding sections in the abstract (p2, lines 44-45), results (p5, lines 102-110), and the discussion (p10, lines 276-278).

Only the BGAT (ABO) and OAS1 proteins should be listed in a new main Table 1, as shown, for example, in Table 1 from Yao et al. <https://pubmed.ncbi.nlm.nih.gov/30111768/>, as the text in lines 108-112 is so short and does not describe this fact that only these two genes/proteins meet the accepted genome-wide colocalization significance criteria from both GWAS summary statistics, as used in other publications. ELF5 can also be mentioned here (possibly also in a main Table 1) as the most promising new candidate without meeting the accepted genome-wide colocalization significance criteria, as shown in Figure 3, with genome-wide significance in the HGI COVID19 data, but unfortunately no pQTL study-related genome-wide significance. All other 5 mentioned "causal" protein candidates (SFTPD, CSF3, HSP40, RAB2A, NUDT5) do not show genome-wide significance in any of the 4 HGI COVID10 GWAS data (sometimes not even $p < 10^{-4}$) nor in the pQTL GWAS data and therefore do not fit the description of "causal candidate" at all, as the locus itself does not show significance for COVID19.

R1 response 5 We agree with the reviewer that the results currently do not justify calling CSF3, HSP40, NUDT5, and RAB2A causal candidate genes and have relabelled them as 'suggestive candidate genes/proteins' (e.g., p5, line 107). We created a novel Table 1 as suggested.

Another question: After the previous candidate selection, why doesn't the HINT1 gene (line 12 in Suppl Table 2) qualify as another suggestive gene/protein (instead of HSP40, RAB2A and NUDT5), with a much smaller $P = 2.14 \times 10^{-14}$ for protein GWAS, PP_H4=84.21% and some "more" suggestive HGI COVID GWAS $P = 7.80 \times 10^{-5}$? And what about the FAS gene with protein GWAS $P = 5.92 \times 10^{-131}$, PP_H4=93.07%, HGI COVID GWAS $P = 1.04 \times 10^{-4}$ in Suppl Table2? For clarity, it might be helpful dividing Suppl Table 2 into the 4 coloc sub-result tables for the 4 HGI GWAS sumstats coloc analyses and then sort the results by significance, with significant genes like BGAT and OAS1 at the top.

R1 response 6 Although HINT1 and FAS certainly justify the PP threshold for a shared genetic signal, both missed our second criterion, that is that the respective regional lead variants are in strong LD ($r^2 > 0.8$). While HINT1 was in relatively high LD just below the threshold ($r^2 = 0.79$), the lead signal at the FAS locus for COVID-19 was distinct ($r^2 = 0.12$), as also evident from the figure below. We added this information to the revised manuscript (p15, lines 431-434).

We further followed the suggestion by the reviewer and subdivided Supplemental table 2 into four different tables.

Rebuttal Figure 1 Locuscompare plot at *FAS* for *FAS* measured using SomaScan (*res_invn_X9459_7*) and severe COVID-19 (A2). The plot on the left opposes $-\log_{10}(\text{p-value})$ for SNPs located in a 500kb window around *FAS* and variants are coloured based on linkage disequilibrium with the lead *cis*-pQTL. The plots on the right hand are stacked locuszoom plots for the same association statistics.

For the same reason, all genes except *BGAT*, *OAS1* and *ELF5* should be removed from Figure 4, as there might be statistically "better" candidates such as *HINT1* or *Fas* from Suppl Table 2, which were not mentioned here. *CSF3* in Figure 6 should not (yet) be mentioned as a COVID19 locus risk because not genome-wide significant, with the indication that this locus may yet be detected in the future in an even larger study than that of the HGI to date.

R1 response 7 We agree with the reviewer that our candidate proteins fall into different categories, and we now clearly highlight this distinction in the revised version of the manuscript. The purpose of the lower panel of Figure 4, however, is purely descriptive and refers only to the expression pattern of the protein coding genes across cell types in healthy donors and does not relate to COVID-19 pathology. We think that this level of information is valuable to the broad readership, as, for example, *SFTPD* is a classical lineage marker gene for relevant cell-types, like AT2 cells.

For better reproducibility, please write in the methods section for lines 420-441 how many genetic variants remain after each filter step.

R1 response 8 We thank the reviewer for this helpful comment. Since the number of overlapping SNPs is highly dependent on the genomic region, we now provide this number in the updated Supplemental table 2.

Line 422: For the coloc program analysis, please report which prior probabilities p_1, p_2, p_{12} were used because number of variants with $PP_{H4} > 0.8$ depends also on the value of prior p_{12} . Please indicate from where the MAFs originate, which were used as input per SNP. It has been shown that the default value of $1e-5$ for the p_{12} prior of the coloc program shifts protein-phenotype pairs often too easily above the threshold of 0.8. (Supplementary Fig. 12 (A); Gudjonsson et al; <https://pubmed.ncbi.nlm.nih.gov/35078996/>). Please repeat also the

analysis with a more stringent prior selection, $p_{12} = 5 \times 10^{-6}$, as recently proposed (Wallace 2020 <https://pubmed.ncbi.nlm.nih.gov/32310995/>) and as performed in Gudjonsson et al., since $p_{12} = 10^{-5}$ is overly liberal (Wallace 2020). Please use prior probabilities of COVID19 case-control association e.g. $p_1 = 23/2000000 = 1.15 \times 10^{-5}$ (i.e. number of gws case-control hits / number of common SNPs (about 2,000,000), see rules from Wallace 2020; i.e. currently 23 gws loci from COVID19 HGI sumstats). Prior p_2 should perhaps also be set at 1.15×10^{-5} rather than 1×10^{-4} , because for SNP-protein GWAS, the study-based genome-wide significance threshold is far below that of case-control GWAS because there are thousands of protein tests genome-wide, e.g., 1.004×10^{-11} in Pietzner 2021 Science and 1.046×10^{-11} in Gudjonsson et al.

R1 response 9 We agree with the reviewer that prior settings can have a strong influence on colocalization results and implemented a sensitivity analysis as explained above. We have now also updated the methods section to include the prior settings used for the previous analysis (p14-15, lines 426-428).

Please make all pQTL summary statistics mentioned in lines 401-411 and used for colocalization analysis publicly available for download with a link in Section "DATA AVAILABILITY" (line 536-541).

R1 response 10 We have now provided a download link in the corresponding section as suggested.

Can the authors please provide quantile-quantile plots and inflation factor calculations for the pQTL GWAS summary statistics used here? (because suggestive signals were examined) (Similar to the publication by Gudjonsson and colleagues <https://pubmed.ncbi.nlm.nih.gov/35078996/>)

R1 response 11 We agree that quantile-quantile plots and inflation factors are standard QC metrics for genome-wide association statistics and can guide identification of spurious results. However, our focus on protein coding regions rather than genome-wide associations breaks with the assumptions made for such QC measures, since they do not represent a random sample of the genome. For example, the computation of genomic inflation factors assumes that at least half of the genome is not associated with the trait of interest and hence association statistics of cis-pQTL regions would be categorized as highly inflated.

Line 435: The number "1,245 unique protein targets with trustworthy colocalization results" cannot be identified or counted (when counting proteins) in Suppl Table 2.

R1 response 12 We agree that this needed clarification and have introduced a common protein name across the two proteomic platforms, covering a total of 1,121 unique proteins after harmonization. We have also clarified this in the methods (p15, lines 443).

Regarding lines 436-441: Please add the different SNP variant IDs for the 4 HGI COVID-19 sum statistics to show the best proxy SNP for each of the COVID19 sum statistics colocalized with the best SNP from the protein GWAS. Please also include LD values between all SNP-SNP pairs from individual protein and COVID-19 sum comparisons. Otherwise, results are unfortunately

not reproducible or usable for other groups. The authors can be guided in the preparation of Suppl Table 2 by the Data Table 10 by Gudjonsson and colleagues, see <https://www.nature.com/articles/s41467-021-27850-z#Sec18>

R1 response 13 *We followed the recommendation of the reviewer and report on top of the LD between regional lead variants (which was a criterion to identify candidate proteins), all requested variants and LD-values in the updated Supplementary Tables 2a-d. We flagged regions, for which some of LD information based on the Fenland or 1000G data sets were not available. We were, unfortunately, unable to identify LD comparisons in the Supplemental tables from Gudjonsson et al. but hope that the current presentation meets the request by the reviewer.*

The sentence in line 432 is not clear to me, please specify: “we further tracked whether the lead cis-pQTL was sufficiently preserved in the set of SNPs common to the protein and COVID-19 summary statistics”. Please explain what this means.

R1 response 14 *We agree that our wording was confusing and have clarified the corresponding section in the revised manuscript (p15, lines 439-440). Briefly, we tested whether the lead cis-pQTL or a proxy in strong LD ($r^2 > 0.8$) within the cis window was included in the overlapping set of SNPs used for colocalization with COVID-19 statistics to ensure valid inference based on the lead cis-pQTL. The fact that only half of the potentially testable proteins survived this filter greatly emphasizes this important step, which has been rarely implemented in other studies.*

Unfortunately, the answer "R1 response 11" is not clear to me. Why are none of the signals from Figure 2 at least nominally significant and replicable across the two pQTLs GWAS summary statistics? Is there at least the same trend in terms of effect sizes? How to be sure that they are not false-positive pQTL associations then? Do the pQTL signals come only from one pQTL sumstats file but not from the other? Please explain.

R1 response 15 *The reviewer highlights an important issue in pGWAS studies based on different underlying measurement technologies. We have previously studied this extensively and shown (Pietzner et al Nat Comms 2021) that assays that supposedly target the same protein can have distinct cis-pQTLs in the same genetic region and that colocalization with disease outcomes or other clinical phenotypes greatly increases the probability to distinguish genuine pQTLs from potential assay specific binding effects.*

For example, we identified two distinct lead cis-pQTLs for SFTPD for SomaScan (rs11201057) and Olink (rs721917) both of which are only in weak LD ($r^2 = 0.142$; see figure below). Only the lead cis-pQTL for Olink, but not the lead cis-pQTL for SomaScan was shared with the risk for severe COVID-19. This might be best explained by the fact, that rs721917 is a missense variant (p.Met31Thr), that, besides its potential functional impact, might also affect recognition of SFTPD by the Olink reagent ($\beta = -0.66$, $p = 5.3 \times 10^{-31}$) but way less so for the SomaScan reagent ($\beta = 0.08$, $p = 3.2 \times 10^{-10}$). A similar reason likely applies to G-CSF as well. Such a complexity has been the reason why we provide platform annotation for protein targets and basically treated protein – platform combinations as unique entities, that is, testing for colocalization

between summary statistics for both protein assays with COVID-19 separately. We clarified this in the revised method section (p15, lines 437-438).

Rebuttal Figure 2 Stacked locuszoom plot at *SFTPD* contrasting the associations seen with SomaScan (bottom) and Olink (middle) and possible alignment with COVID-19 (top). Marginal association statistics for protein levels and COVID-19 are shown for a 500kb region around *SFTPD*. Single variants have been coloured according to their linkage disequilibrium with the respective lead cis-pQTLs for SomaScan (rs11201057, blue) and Olink (rs721917, orange). The lines in the bottom panel show the location of each genetic signal.

Regarding “R1 response 11”: If the P-values in Figure 2 are from Mendelian randomization analysis, the p-value threshold for determining significance must be defined somewhere. To me it looks more like the P-values shown in Figure 2 (contrary to the Figure 2 description) are not MR P-values but the P-values from the HGI COVID19 SNP Sumstats, see also Supp Table 2, columns "Pvalue". The same P-values from Figure 2 are also listed in Suppl Table 3 as "pval.mr" (legend for column names is missing). I have looked in the HIG COVID19 data myself and these are all HIG COVID19 SNP P-values and therefore cannot be MR P-values. So is there something wrong with the P-values of the MR analyses provided in Figure 2 and Suppl Table

3? Can the authors please check once the MR P values with another alternative method, e.g. Summary-data-based Mendelian Randomization tool SMR (Zhu 2016 Nat Genet. PubmedID <https://pubmed.ncbi.nlm.nih.gov/27019110/>) or MR-Egger (Bowden 2015 Int J Epidemiol <https://pubmed.ncbi.nlm.nih.gov/26050253/>) since I am unfortunately not familiar with the Wald ratio MR method of Lawlor 2008, which is mentioned only by two sentences in the method section?

R1 response 16 We thank the reviewer for pointing out this important weakness in the description of our statistical approach and rephrased the corresponding section on MR (p16, lines 472-475). We have used the simplest, and indeed in our case only valid MR method, that is, a simple ratio estimate between the effect of the SNP on the exposure (protein) and the outcome (COVID-19). This is the fundamental basis of any other two-sample MR method (Burgess et al. 2013 Genet. Epidemiol.). P-values for the effect estimate then incorporate uncertainty in the outcome measurement only and hence are identical with what is reported for the outcome GWAS (Burgess et al. 2013 Genet. Epidemiol.). It is a direct consequence of the 'one causal variant' assumption made during colocalization. As the reviewer already pointed out, most of the presented cis-pQTLs did not meet the corrected genome-wide statistical significance, let alone the identification of additional variants that would allow for the use of more sophisticated MR methods such as (G)SMR. We note that an MR approach like ours has been successfully implemented previously in linking proteins to COVID-19 (Zhou et al. 2021 Nature Medicine, Gaziano et al. 2021 Nature Medicine) or more broadly to diseases (Zheng et al. 2020 Nature Genetics).

Minor points

Regarding the answer "R1 answer 4", the text in lines 133-134 still lacks the clear indication that the pQTL SNP rs766826 has also already been identified as eQTL SNP rs766826 by COVID19 HGI publication #15, which of course should not be a requirement here, but further emphasizes the validation at the protein level here.

R1 response 17 We clarified this in the revised version of the manuscript (p6, line 134).

New Lines 112/113: "All genetic variants possibly linking proteins to COVID-19 had highest effect estimates for severe COVID-19". Which effect estimates of which analysis are meant here? From the case-control effect estimates of the HGI GWAS data (Suppl Table 3) or are we already referring to the Mendelian randomization analysis (also results in Suppl Table 3)? If the former, then please provide case-control effect estimates of SNPs in ORs in Suppl Table 3 (and also new main Table 1), otherwise it is difficult to understand the statement "highest effect estimates" when showing beta values instead of ORs for case-control analyses.

R1 response 17 We thank the reviewer for pointing out this ambiguity and have now clarified the corresponding section in the revised version of the manuscript (p12, line 349). Briefly SNP – outcome effects on the log(odds) scale, that do not take protein levels into account, differed substantially across the four outcome definitions, which is an approach comparable to what the COVID-19 HGI did for locus classification. We provide the relevant data in the new Supplementary Table 4

REVIEWERS' COMMENTS

Reviewer #1 (Remarks to the Author):

Thank you for the detailed answers to my questions and the explanatory figures. If the authors please still fulfill these questions below, especially the missing protein GWAS summary statistics, then I have no further concerns.

Major Points

Regarding R1 response 10:

Unfortunately, I cannot download the protein GWAS summary statistics promised in the response via the download link <https://omicscience.org/apps/pgwas/> provided in the DATA AVAILABILITY section. The following text is displayed on the web page: "Data access: We are currently working on an application process to provide access to genome-wide summary statistics for scientific use." Please provide a link to the full summary statistics from all SomaScan and Olink pQTL GWAS as compressed files immediately and before publication. Providing summary statistics for the research community is not a problem from a data protection point of view, but it allows for reproducibility to a great extent and is also a valuable resource for the entire research community.

Regarding R1 response 1:

I find the idea of 3 tiers very helpful. Since ELF5 from tier 2 is neither genome-wide significant after adjusting for the number of proteins tested in the protein GWAS, nor has unadjusted genome-wide significance ($P < 5 \times 10^{-8}$) like ABO and OASL1 from tier 1 (Table 1) in the protein GWAS, I would suggest referring to ELF5 in the manuscript title only as a "candidate risk gene."

Minor points

Regarding R1 response 3:

In the new supplementary table 3, which is referred to in lines 108-109 and line 428, no results concerning "grid of prior combinations" are visible. Must be supplementary table 4.

REVIEWERS' COMMENTS

Reviewer #1 (Remarks to the Author):

Thank you for the detailed answers to my questions and the explanatory figures. If the authors please still fulfill these questions below, especially the missing protein GWAS summary statistics, then I have no further concerns.

Major Points

Regarding R1 response 10:

Unfortunately, I cannot download the protein GWAS summary statistics promised in the response via the download link <https://omicscience.org/apps/pgwas/> provided in the DATA AVAILABILITY section. The following text is displayed on the web page: "Data access: We are currently working on an application process to provide access to genome-wide summary statistics for scientific use." Please provide a link to the full summary statistics from all SomaScan and Olink pQTL GWAS as compressed files immediately and before publication. Providing summary statistics for the research community is not a problem from a data protection point of view, but it allows for reproducibility to a great extent and is also a valuable resource for the entire research community.

We apologize for the delay in providing the summary statistics and have now implemented a flexible framework to download all or specific protein summary statistics as a resource for the research community - <https://omicscience.org/apps/pgwas/> (SomaScan v4) and <https://zenodo.org/record/6787142#.Yr761uxBxhE> (Olink).

Regarding R1 response 1:

I find the idea of 3 tiers very helpful. Since ELF5 from tier 2 is neither genome-wide significant after adjusting for the number of proteins tested in the protein GWAS, nor has unadjusted genome-wide significance ($P < 5 \times 10^{-8}$) like ABO and OASL1 from tier 1 (Table 1) in the protein GWAS, I would suggest referring to ELF5 in the manuscript title only as a "candidate risk gene."

While the lead variant mapping to ELF5 does not pass genome-wide significance in Table 1, it does in the meta-analysis by the COVID-19 HGI ($p = 2.93 \times 10^{-12}$), which also included 23&me, and has further been validated by several other groups, including GenOMICC, as outlined in the main text (p5/6, lines 130-133 and p12, line 366). We therefore think that the cumulative evidence provided by us, and others justifies calling ELF5 a risk gene beyond candidate status.

Minor points

Regarding R1 response 3:

In the new supplementary table 3, which is referred to in lines 108-109 and line 428, no results concerning "grid of prior combinations" are visible. Must be supplementary table 4.

We thank the reviewer for this advice and corrected the reference to table ST4 accordingly (now Supplementary Data 6).